# Medium-scale integrated circuits based on p-type 2D semiconducting MoTe$_2$

Hui Wang[1], Zebang Luo[1], Biyuan Zheng [1], Zilan Tang[1], Huaidong Ye[2], Yulong Yuan[2], Yizhe Wang[1], Haitao Zhang[1], Qin Shuai[1], Huawei Liu[1], Guangcheng Wu[1], Dong Li [1], Li Xiang [1] ✉ & Anlian Pan [3] ✉

Two-dimensional (2D) semiconductors hold promise for next-generation electronics, yet the lack of scalable p-type counterparts remains a major bottleneck, with prior studies largely limited to discrete devices or simple circuits. Here we report the realization of medium-scale integrated circuits (MSICs) based on wafer-scale p-type 2D semiconductors, enabled by the controlled synthesis of uniform 4-inch 2H-MoTe$_2$ films. A precursor-engineering strategy that integrates thickness-tunable Mo precursors with a sustained-release chalcogen supply enables deterministic thickness control and wafer-scale uniformity. The resulting p-type transistors exhibit highly reproducible characteristics, including on/off ratios of ~10$^5$ and mobilities of ~7 cm$^2$ V$^{-1}$ s$^{-1}$ under low operating voltages. Leveraging a device density exceeding 1300 cm$^{-2}$, we demonstrate a 140-transistor full adder, showing the potential of our approach towards the realization of future large-scale 2D complementary metal-oxide-semiconductor (CMOS) circuits.

The relentless scaling of silicon transistors now faces fundamental or fabrication limits, driving urgent exploration of alternative channel materials for next-generation integrated circuits (ICs)[1–3]. Among emerging candidates, two-dimensional (2D) transition metal dichalcogenides (TMDs) have emerged as promising materials due to their dangling-bond-free surfaces[4] superior electrostatic control[5,6] and generous band structures[7,8], offering improved device performance, energy efficiency and circuit functionality[1,7,9]. While n-type 2D TMDs (e.g. the MoS$_2$)[4–7,10–14] have achieved wafer-scale synthesis[13,14] (up to 12-inch), high-performance device construction[12,15] and functional circuit integration with over 1,000 transistors[12–16], their p-type counterparts remain trapped in a discrete device level demonstrations despite years of efforts.

Recent progress in metal-organic chemical vapor deposition (MOCVD) has enabled 2-inch tungsten diselenide (WSe$_2$) epitaxy[14,15], and tellurization strategies produced 2H-MoTe$_2$ films with promising individual-device metrics[17–19]. However, as with all p-type 2D semiconductors, there exists a substantial gap between the emerging demonstration–be integrated with only several discrete devices (or even a single device)[8,20,21]–and industrial ICs that comprise at least hundreds of transistors. From the standpoint of integrated circuitry, there are three fundamental challenges that need to be simultaneously addressed for transitioning the p-type 2D devices from proof-of-concept devices to functional ICs.

(i) Manufacturability: Current p-type 2D-TMD demonstrations (e.g., WSe$_2$, MoTe$_2$) are typically confined to dimensions of only a few millimeters or centimeters, which is impractical and ineffective for industry standards[8,20–26]. Within the IC industry, 2D semiconductors must be scaled up to match industry-standard substrate sizes of 100 mm or larger in diameter[1,9,13], thereby enabling high-throughput manufacturing processes that are economically viable and scalable for commercial applications.

(ii) Device performance: While p-type transistors fabricated from mechanically exfoliated or single-domain-sized 2D TMD flakes have demonstrated exceptional electrical characteristics, their wafer-scale-synthesized counterparts typically suffer from compromised performance metrics, including limited current on-off

[1]Key Laboratory for Micro-Nano Physics and Technology of Hunan Province, Hunan Institute of Optoelectronic Integration, College of Materials Science and Engineering, Hunan University, Changsha, China. [2]Hunan Institute of Advanced Sensing and Information Technology, Xiangtan University, Xiangtan, China. [3]School of Physics and Electronics, Hunan Normal University, Changsha, China. ✉e-mail: xiangli93@hnu.edu.cn; anlian.pan@hnu.edu.cn

ratios and high operating voltages (generally reaching tens of volts). This performance gap is primarily attributed to insufficient electrostatic gate control in these p-type 2D transistors[21,26,27]. These characteristics suggest more intensive efforts are necessary in device engineering to refine the large-scale 2D transistors, such as scaling down the thickness of the 2D semiconducting channel and the oxide thickness.

(iii) Uniformity and integration density: Previous attempts to integrate 2D TMDs into circuits have frequently utilized materials that are not air-stable[5] or have had to rely on fabrication processes incompatible with commercial industrial facilities (e.g., the transfer printing of contacts and dielectrics)[25,28,29]. These factors would lead to substantial challenges in process control, robustness, and reproducibility of p-type 2D semiconducting devices, thereby significantly hindering their potential for large-scale and high-density integration (present 2D circuits utilize p-type transistors are in the scale of only a few discrete devices[8,20–26]).

In this article, we address these challenges through a co-optimization approach: (i) Controlled Material synthesis: sustained-release Te precursor engineering combined with ultrathin metal precursor deposition enables 4-inch 2H-MoTe$_2$ growth with great uniformity and thickness-controllability. (ii) Device engineering: Scaling down the semiconductor and oxide thickness to enhance the electrostatic control, enabling p-type 2D transistors with large current on/off ratios of $(1.29 \pm 0.36) \times 10^5$ and 4 V operation voltages. (iii) Utilizing highly-controllable fabrication processes to yields 1346 transistors/cm$^2$ device density and performance uniformity (standard deviation in threshold voltage of 42.71 mV). The achieved uniformity parameters are superior to those reported for most 2D transistors and even comparable to those of state-of-the-art silicon-based transistors in foundries. In summary, the key advances of this material growth are the scaling of uniform 2H-MoTe$_2$ to a 4-inch wafer and the precise thinning of the channel down to 3 layers. Together, these achievements enable the medium-scale integration of more than 100 p-type 2D transistors, with significantly improved device performance. These demonstrations effectively bridge the gap between current, discrete device-level demonstrations and practical mass device integration, validating the utility of p-type 2D semiconductors in medium-scale integrated circuits comprising hundreds of transistors. Such results will lay a solid foundation for the future large-scale integration of 2D CMOS circuits, further advancing the progress of 2D semiconductors towards industrial applications.

## Results

### 4-inch uniform synthesis of 2H-MoTe$_2$ via precursor engineering

Achieving uniform growth of 2D semiconductors on large-scale wafers requires overcoming fundamental limitations in precursor dynamics[13,30]. Conventional chemical vapor deposition (CVD) systems suffer from temporal and spatial fluctuations in chalcogen precursor supply due to rapid sublimation and low vapor pressure of solid powders, leading to non-uniform nucleation[30]. Our custom-designed CVD systems (shown in Fig. 1a and Supplementary Fig. 1) integrate two key metrics: (i)Sustained-release tellurium delivery through tellurium source resolidification and molecular sieve envelopment; and (ii) ultrathin Mo precursor deposition enabled by oxygen plasma-activated substrate engineering.

The tellurium precursor management system (Fig. 1b) addresses the critical challenge of maintaining a stable and controlled supply across wafer-scale growth. By melting Te powder at 450 °C followed by natural cooling recondensation, a more stable Te flux could be realized compared to conventional powder sublimation methods. According to previous TMD growth reports, chalcogen precursors, in comparison to metal precursors, exhibit significantly higher melting points and a

slower sublimation rate[30]. This difference can result in excessive nucleation and the formation of localized chalcogen vacancies within the synthesized 2D TMDs. Therefore, we envelop molecular sieves on the resolidified Te (Fig.1b), which could ensure stoichiometric Te:Mo ratio during the wafer-scale growth by directing the tellurium vapor release exclusively through the sieve's pores, thereby precisely control the precursor evaporation rate and suppresses Te vacancy formation. Concurrently, the metal precursor thickness control was also critical during MoTe$_2$ growth, which would have a significant influence on the resulting thickness of the MoTe$_2$ films. Despite recent advances that have facilitated the wafer-scale growth of 2H-MoTe$_2$[18,19,26], the thinning of the resultant MoTe$_2$ to the necessary dimensions remains challenging. Previous reports have typically yielded films with thicknesses exceeding six layers or even higher[18,26,27,31,32], which would ultimately limit their electrostatic control capabilities in high-performance transistors. Generally, the direct deposition of metal precursors often results in island-shaped and discontinuous structures according to the Volmer-Weber model[33], which sets unavoidable barriers for the growth of ultrathin continuous MoTe$_2$ films. We overcome this Volmer-Weber limitation for ultrathin Mo precursors through substrate surface modification. High-energy oxygen plasma treatment could enhance the wettability and adhesion of the Mo precursor on the substrates. This treatment changed the substrate surface energy, endowing it with hydrophilic properties (Supplementary Fig. 2), and activates the surface with a greater number of active sites, thereby enabling the deposition of an ultrathin Mo film of 2 nm (Fig. 1c). Furthermore, the enhanced adhesion resulting from the oxygen plasma treatment also effectively suppress Mo islanding and evaporation during high-temperature tellurization, ensuring more consistent and uniform film growth processes. Following the meticulous preparation of the precursors, the homogeneous growth of 4-inch MoTe2 was achieved within our CVD system, and detailed CVD growth procedures are presented in the Method section.

The synergy of these precursor engineering yields 4-inch 2H-MoTe$_2$ films with high uniformity. High-angle annular dark-field scanning transmission electron microscopy (HAADF-STEM) cross-sectional analysis (Fig. 1d) has confirmed the growth of the wafer-scale 2H-MoTe$_2$ film with 2D-layered structures, and the elemental mapping in Fig. 1d identifies the presence of molybdenum and tellurium. In addition, as present in Supplementary Fig. 3, X-ray photoelectron spectroscopy (XPS) spectra across of the synthesized MoTe$_2$ film was analyzed, which also confirmed the presence of molybdenum and tellurium. The high-resolution XPS peaks were observed at 228.6 eV (Mo $3d_{5/2}$), 231.8 eV (Mo $3d_{3/2}$), 573.2 eV (Te $3d_{5/2}$), and 583.6 eV (Te $3d_{3/2}$), respectively, which are also consistent with the characteristics of 2H-MoTe$_2$ in previous reports[18,26,27]. To elucidate the growth mechanism during tellurization, we find that the formation of MoTe$_2$ does not proceed via direct crystallization into the 2H phase. Instead, a metastable 1 T' phase forms at the early stage and subsequently transforms into the thermodynamically stable 2H phase with continued Te supply, with the coexistence of both phases observed during growth providing direct experimental evidence for this phase-transition pathway (Supplementary Fig. 4). As inferred from Supplementary Fig. 4, the largest continuous crystalline domain reaches approximately 900 μm, approaching the millimeter scale. Consequently, despite being polycrystalline, the film's large crystalline domains result in a substantially reduced density of grain boundaries, with typically only one or two intersecting a transistor channel. This microstructural feature is a critical factor enabling uniform device performance and the successful fabrication of medium-scale integrated circuits based on MoTe$_2$. Owing to the controlled-thickness pretreatment of Mo metal precursor, we achieved precise thickness regulation of the resulting MoTe$_2$ films from 3 to 20 layers, respectively, which is present in the optical images and corresponding AFM height profiles of Supplementary Fig. 5. Such layer number control capability directly correlates

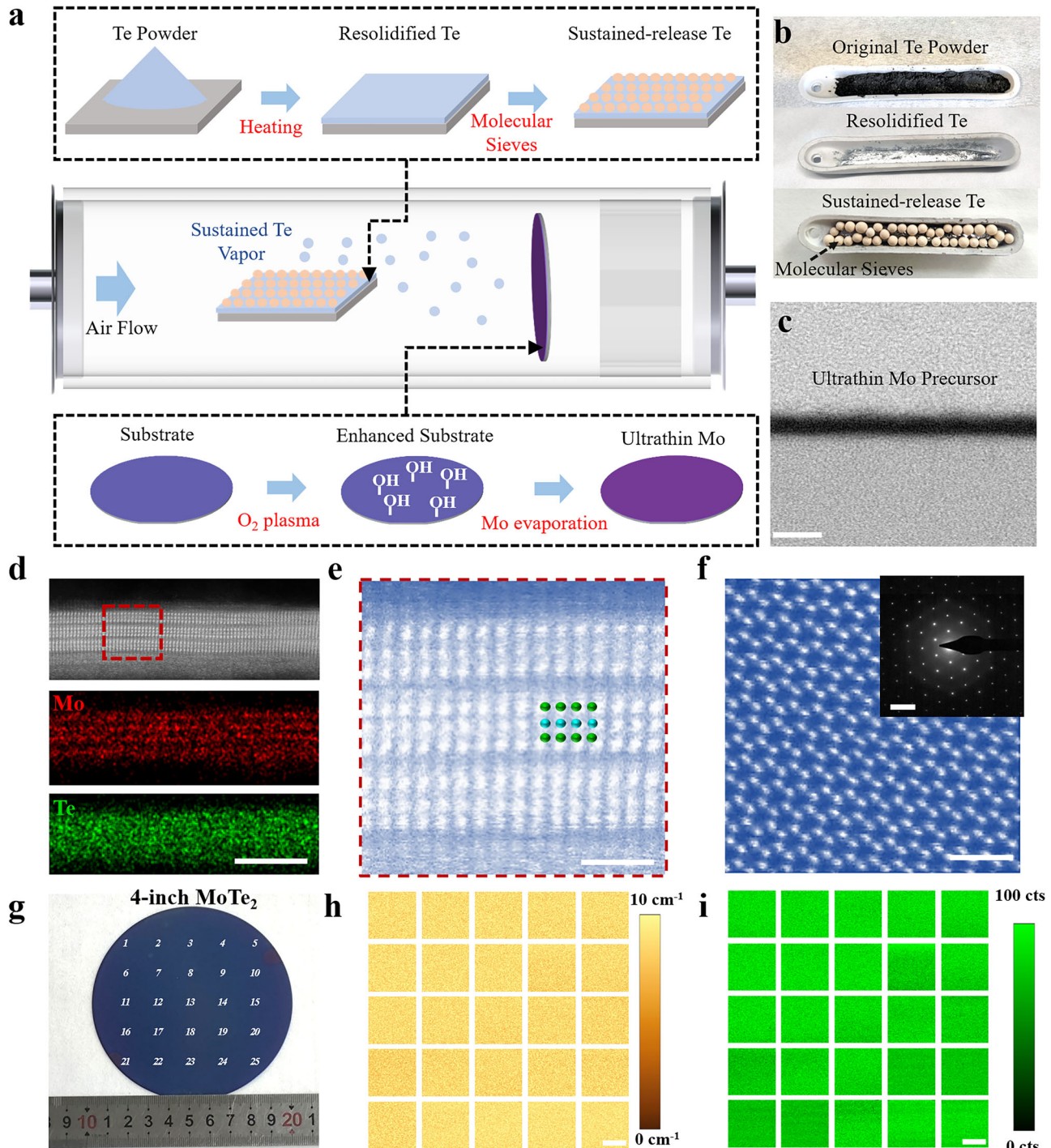

**Fig. 1 | Four-inch wafer-scale growth of MoTe₂ thin films. a** Schematic illustration of the MoTe₂ growth process by precursor engineering. **b** Optical photograph of the original Te powder (top), the resolidified Te (middle), and the sustained-release Te precursor (bottom). **c** Cross-sectional high-angle annular dark-field scanning transmission electron microscopy (HAADF-STEM) image of ultra-thin molybdenum film precursor for MoTe₂ growth. (Scale bar 5 nm). **d** The cross-sectional HAADF-STEM image (top) and the element analysis (middle and bottom) of the prepared 3-layered MoTe₂ thin films. (Scale bar 5 nm). **e** Atomic-resolution HAADF-STEM images of the 3-layered MoTe₂ thin films. (Scale bar 5 nm). **f** Structure of the 3-layered MoTe₂ viewed down its *c* axis with atomic-resolution STEM image, inset is the selected-area electron diffraction (SAED) pattern of the MoTe₂ films selected using an aperture with a diameter of 200 nm. (Scale bar 5 nm). **g** Photograph of the MoTe₂ film grown on a 4-inch wafer with 25 sampling spots. **h**. Spatial Raman peak mapping for the sampling spots, where the color bar represents the full width at half maximum (FWHM) of the Raman peak. (Scale bar 25 μm) **i** Spatial intensity mapping of the $E_{2g}$ Raman mode (234 cm⁻¹) for the sampling spots, where the color bar represents the Raman peak intensity in arbitrary units (a.u.). (Scale bar 25 μm).

with the precursor's initial thickness, enabling predictable synthesis of the 2H-MoTe₂ with designed dimensionality. In addition, the synthesized 2H-MoTe₂ exhibits great thickness uniformity and macroscopic smoothness (surface roughness <1.15 Å), as evidenced by large-area cross-sectional STEM imaging (in Supplementary Fig. 6), which

demonstrates good thickness control during the material synthesis. The amplified cross-sectional STEM image of Fig. 1e reveals a uniform A-B stacking sequence across the entire film with a well-ordered crystalline structure. Meanwhile, the absence of atomic intermixing or defects could also be demonstrated in Fig. 1e. As shown in Fig. 1f, STEM

analysis along the c-axis of MoTe$_2$ was conducted, which presents a highly-crystalline 2H-MoTe$_2$ lattice with very few defects, such as the adatoms and vacancies. The selected-area electron diffraction (SAED) pattern presented in the inset of Fig. 1f further conformed the uniform highly-crystalline characteristics of the obtained MoTe$_2$ thin films. To further assess the structural uniformity and crystallinity of the films at the wafer scale, additional transmission electron microscopy (TEM) characterizations were performed at multiple spatial locations across the MoTe$_2$ films, and the corresponding selected-area electron diffraction (SAED) patterns were obtained (Supplementary Fig. 7). The SAED patterns collected from different regions exhibit well-defined and consistent diffraction spots, indicating good crystallinity and a uniform crystal structure across the wafer. Notably, the 2H-MoTe$_2$ growth could also achieve full coverage on a 4-inch wafer, which is demonstrated in Fig. 1g. To characterize the crystalline quality and homogeneity of the synthesized 2H-MoTe$_2$ thin films, a comprehensive Raman spectroscopic analysis was performed at 25 equidistant points across the 4-inch wafer, as demonstrated and numbered in Fig. 1g. The Raman spectra in Supplementary Fig. 8 corroborate the growth of 2H-MoTe$_2$, with distinct vibrational modes of $A_{1g}$ (171 cm$^{-1}$), $B_{2g}^1$ (291 cm$^{-1}$) and $E_{2g}^1$ (234 cm$^{-1}$) consistently observed across all sampled points, which are consistent with previous reports[18,19,27]. In pursuit of a better understanding of the quality of the 2H-MoTe2 films, Raman mapping was conducted over a 50 μm × 50 μm area at 25 discrete points across the 4-inch wafer, as detailed in Fig.1h and i. The homogenous color distribution within these mappings, coupled with the uniform peak intensity and full width at half maximum (FWHM) of the $E_{2g}^1$ (234 cm$^{-1}$) Raman shift. The statistic standard deviations in peak intensity and FWHM demonstrate very broad distributions of 10.36 ± 2.61 CDD cts and 4.48 ± 0.54 cm$^{-1}$, respectively (with comprehensive statistical histograms provided in Supplementary Fig. 9). Such small distributions demonstrate the great uniformity of the wafer-scale 2H-MoTe$_2$ thin films, not only within a microscale area but also across the entire 4-inch wafer. Therefore, we also conducted additional measurements to more rigorously evaluate the quality and uniformity of the thin film across 4-inch wafers. Atomic force microscopy imaging was performed with scan sizes ranging from 5 × 5 μm$^2$ to 15 × 15 μm$^2$, sampled from multiple locations on the wafer. These images (Supplementary Fig. 10) clearly reveal grain morphology. Concurrently, statistical analysis of surface roughness across 16 representative regions (each 15 × 15 μm$^2$) (Supplementary Fig. 11) confirmed exceptional uniformity and minimal variability. Notably, we also performed large-area scanning electron microscopy (SEM) imaging (Supplementary Fig. 12) to present the micrometre-scale film continuity and topographical features more clearly. This confirms that the 2H-MoTe$_2$ films prepared on wafers exhibit high uniformity on a wafer scale, providing a robust foundation for subsequent transistor and circuit fabrication.

### Electrostatic control via thickness scaling and high-κ

The limitations of wafer-scale p-type 2D transistors, including the limited current on-off ratio (~10$^4$ or lower) and high operating voltages (typically in tens of volts[18,19,27,31]), stem from inadequate electrostatic control within the 2D transistors. Accordingly, we address this through two optimization strategies: atomic-scale channel thinning enabled by precursor engineering and sub-20 nm high-κ dielectric integration.

Firstly, we systematically investigated thickness-dependent characteristics in MoTe$_2$ transistors (device schematic in Fig. 2a). Cross-sectional STEM analysis (Fig. 2b) confirms our CVD-grown 2H- MoTe$_2$ maintains high crystallinity and layered structure from 20 down to 3-layer thickness. Notably, our precursor engineering eliminated conventional CVD growth defects, such as discontinuities and surface contamination, by precisely controlling vapor-phase precursor delivery. In addition, the thickness of the Mo precursor defines the total amount of Mo available for conversion, thereby imposing an intrinsic constraint on the final MoTe$_2$ layer number (Fig. 2c). When the

precursor approaches the ultrathin regime, the growth enters a Mo-limited conversion mode that suppresses excessive nucleation and vertical stacking, enabling the formation of thickness-controlled and uniform few-layer films. Precise layer control of 2H-MoTe$_2$ (3L-20L) via Mo precursor thickness tuning (Fig. 2c) enables linear thickness controllability, which is a significant advancement over previous CVD methods that produced ≥6-layer films[18,19] and achieved the thinnest wafer-scale p-TMD reported (3 layers). Figure 2d compares the transfer characteristics (drain current $I_{ds}$. gate-to-source voltage $V_{gs}$) of MoTe$_2$ transistors with channel thicknesses ranging from 3 to 20 layers, which reveal significant thickness dependence: reducing channel thickness from 20 L to 3 L enhances on-current 21.8 times (from 0.30 to 6.54 μA) while suppressing off-current 81times (from 12.95 to 0.16 nA) (Fig. 2d). This indicates a three-order-of-magnitude improvement in current on/off ratios ($I_{on}/I_{off}$) (from 23 to 1.47 × 10$^5$), outperforming previously-reported CVD-synthesized MoTe$_2$ transistors (benchmarking in Fig. 2e)[18,19,21,26,27,34–40].

To further enhance electrostatic control in MoTe$_2$ transistors, we engineered gate dielectric architectures by replacing 280-nm SiO$_2$ with high-κ alternatives. The high-κ material HfO$_2$ offers the following core advantages: i). Higher gate capacitance and improved electrostatic control: The high dielectric constant (κ) of HfO$_2$ enables a larger gate capacitance per unit area ($C_{ox}$) for a given physical thickness. As described by the relationship for the subthreshold swing, $SS = \ln(10)(kT/q)(1 + C_{it}/C_{ox})$, a larger $C_{ox}$ directly leads to a smaller SS, signifying a steeper switching characteristic. This enhanced gate control is crucial for achieving low-power operation in transistors. ii). Scaled equivalent oxide thickness (EOT): The high κ value allows for the realization of a very small EOT while maintaining a physically thicker dielectric layer. This combination is key to enhancing gate control efficiency while effectively suppressing the gate leakage current that becomes prohibitive with ultra-thin conventional dielectrics. In this study, we demonstrate that the ALD-grown HfO$_2$ dielectric, with higher κ compared to SiO$_2$ and Al$_2$O$_3$, provides superior electrostatic control over the MoTe$_2$ channel. Figure 2f illustrates devices integrating three-layer MoTe$_2$ channels with three dielectric configurations: 280-nm SiO$_2$, 20-nm Al$_2$O$_3$, and 16-nm HfO$_2$. The corresponding transfer characteristics are detailed in Fig. 2g. Transistors with SiO$_2$ dielectrics required 150 V gate swings (−100 V to +50 V) for full on-off switching (shown in Fig. 2d and 2g), which is a voltage regime incompatible with low-power electronics. Leveraging high-κ materials (Al$_2$O$_3$, HfO$_2$) validated in silicon CMOS technologies, we achieved 37.5× voltage reduction (4 V operation at −4 V to 0 V) while maintaining $I_{on}/I_{off}$ of beyond 10$^5$ (Fig. 2g). As Fig. 2h illustrates, the MoTe$_2$ devices achieve a combination of low-voltage operation (4 V) and high current on/off ratios (1.47 × 10$^5$), demonstrating operational superiority against the previously reported CVD-synthesized counterparts[18,19,21,26,27,34–40]. Although a local back-gate configuration is adopted in this study to preserve channel quality, top-gate architectures remain important for achieving high-density integration in two-dimensional electronics. However, the direct deposition of high-κ dielectrics on two-dimensional semiconductors presents intrinsic interfacial challenges. Owing to the absence of dangling bonds, ALD growth is nucleation-limited and can introduce interface defects, thereby degrading carrier transport and device uniformity. Interface engineering strategies, such as the introduction of an ultrathin seed layer, appropriate surface modification, and interfacial treatments[1,9], can promote continuous dielectric growth and reduce interfacial defect density, providing a feasible pathway toward high-quality top-gated device integration.

### Highly uniform MoTe$_2$ transistors for integration

The transition from discrete 2D devices to integrated circuits demands achieving silicon-level uniformity in threshold voltage ($V_{th}$) and on/off ratio, which is a challenge for 2D semiconductors. As depicted in Fig. 3a, our p-type MoTe$_2$ transistors employ a gate-first architecture

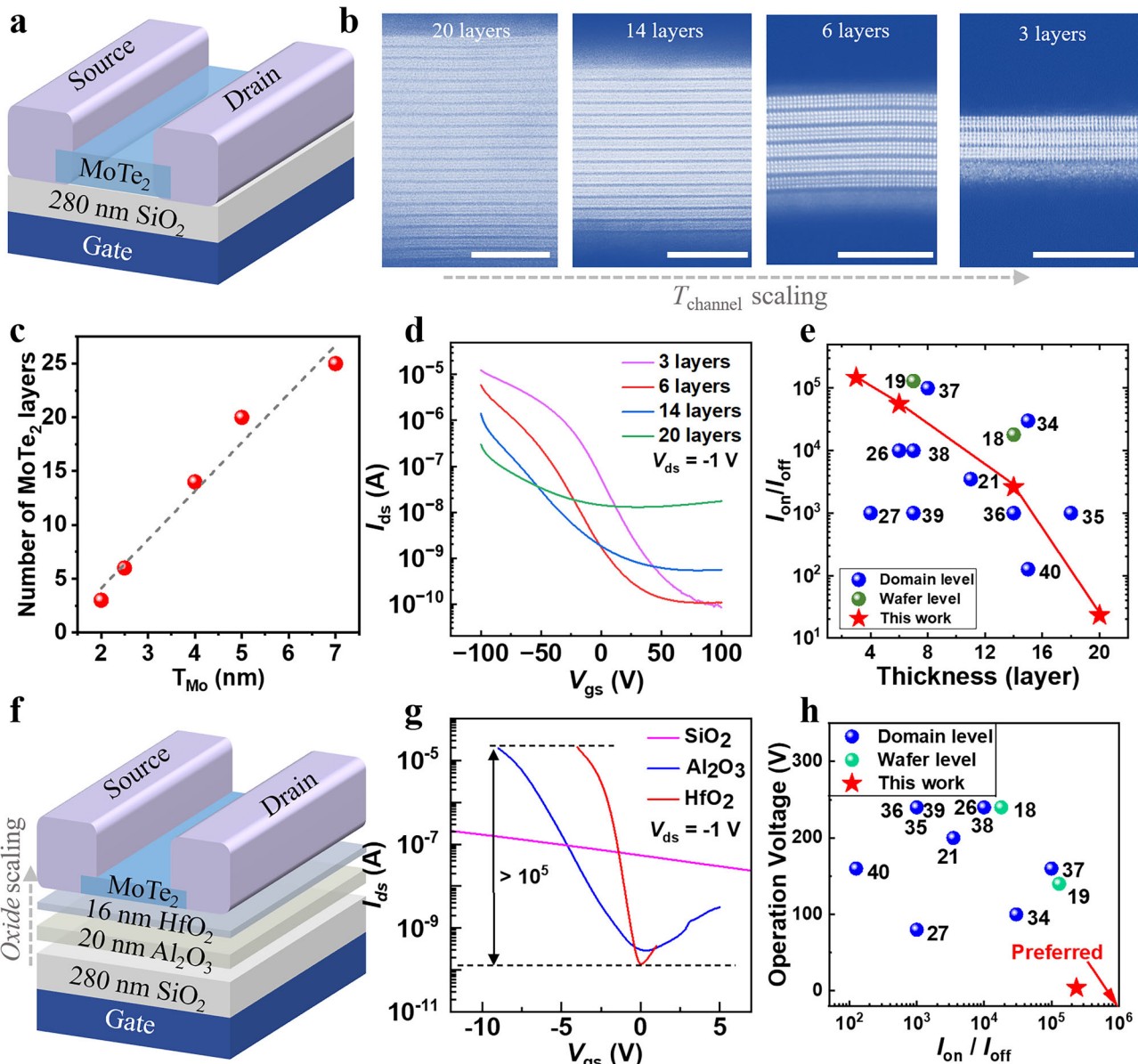

**Fig. 2 | Performance improvements of MoTe₂ transistors by device engineering.** **a** Device structure based on a MoTe₂ film on a dielectric 280-nm-thick SiO₂ substrate. **b** The cross-sectional HAADF-STEM image of the thickness scaling from 20, 14, 6 to 3 layers of MoTe2 thin films. (Scale bar 5 nm). **c** The thickness correspondence curve between the initial Mo precursor and post-reacted MoTe₂. (The dashed line represents the linearity of the fit.) **d** Transfer characteristics of the MoTe₂ transistors with varied channel thickness. (drain current $I_{ds}$. gate-to-source voltage $V_{gs}$. drain-to-source voltage $V_{ds}$.) **e** The performance comparison in terms of $I_{on}/I_{off}$ and thickness correspondence between this work and previous reports[18,19,21,26,27,34–40]. **f** Device structure with a 3-layer-thick of MoTe₂ on three alternative dielectric configurations: 280-nm SiO₂, 20-nm Al₂O₃, or 16-nm HfO₂. **g** Transfer characteristics of the MoTe₂ transistors with varied dielectrics. (The horizontal dashed line marks an on/off ratio of 10⁵ for devices using HfO₂ as the gate dielectric.) **h** The performance comparison in terms of $I_{on}/I_{off}$ and operation voltage window between this work and previous reports[18,19,21,26,27,34–40].

that synergizes two strategic advantages: processes enable precise atomic-layer-deposited (ALD) dielectrics, and minimized thermal/chemical exposure (<3 processing steps) to the 2D channel post-transfer. It is noteworthy that the material transfer procedure is employed solely once to transfer the MoTe₂ films from the growth substrate to pre-patterned substrates 5-nm Ti/20-nm Au local gates and 16-nm HfO₂ high-κ dielectrics. For this reason, we intentionally adopted the bottom-gate configuration to ensure a high-quality MoTe2/dielectric interface and to enable reliable fundamental transistor functions without the effects of a non-ideal top dielectric interface. This approach allows us to present a baseline performance that reflects the material's potential under well-controlled manufacturing processes, particularly for evaluating scalability in large-scale circuit

integration. The remaining components of the MoTe₂ transistors utilize the standard semiconductor manufacturing facilities, with the objective of making the entire device fabrication process as controllable and reproducible as feasible. Concurrently, the transfer stamp, fabricated from thermal release tape (TRT) and polymethyl methacrylate (PMMA) (for detailed procedures, see Methods), in conjunction with heat pressing, has been proven to be reliable for single-layer nanomaterials transferring[41,42]. Our wafer-scale integration strategy enables high-density fabrication of MoTe₂ transistor arrays and functional circuits across 4-inch substrates (Fig. 3b), achieving a device integration density of over $1.3 \times 10^3 \, cm^{-2}$. In comparison to previously reported p-type 2D transistors[18,19,24,31], this represents a great improvement over prior p-type 2D transistor platforms, and

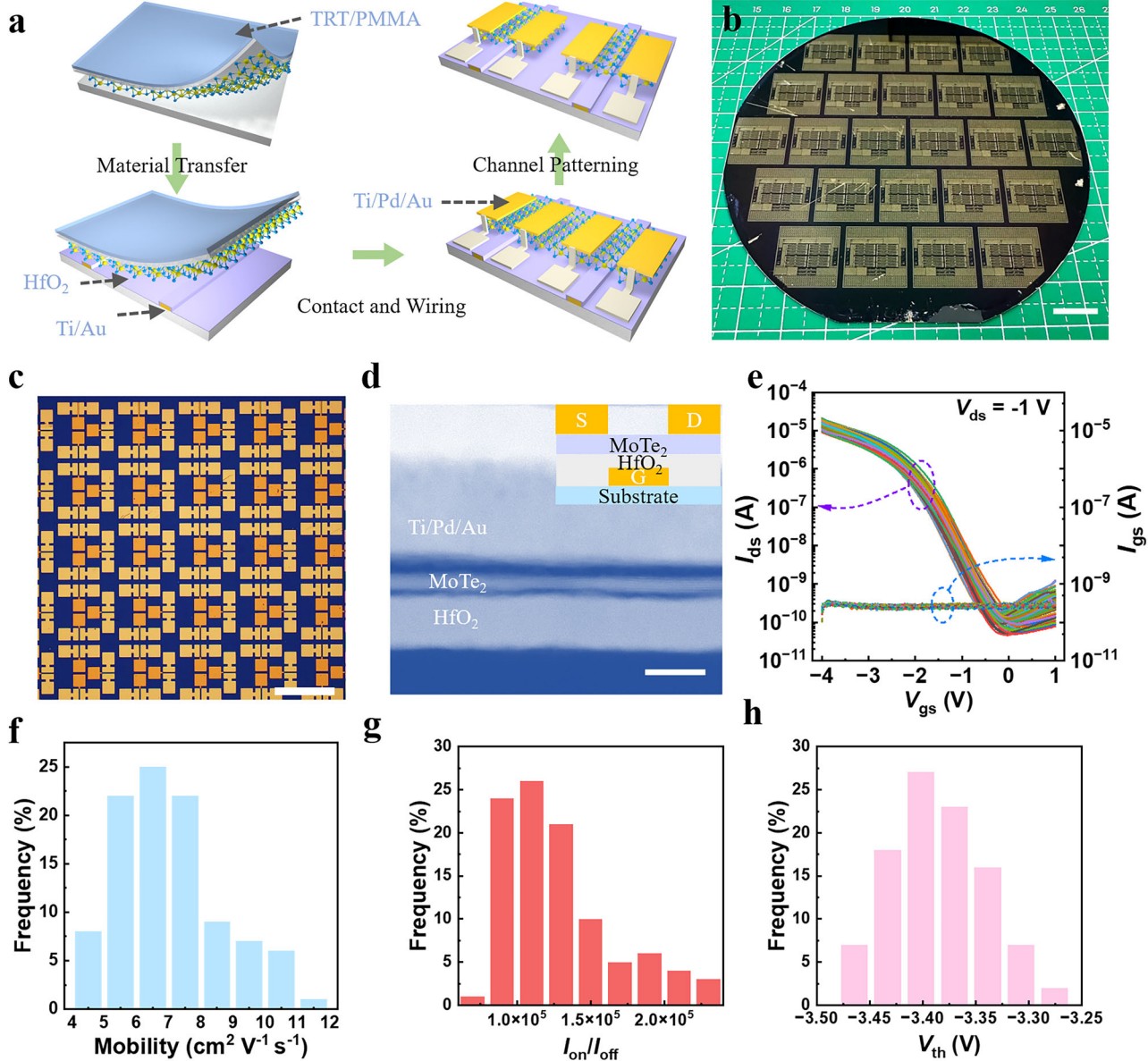

**Fig. 3 | Highly-uniform p-type MoTe₂ transistors array. a** The fabrication process of the p-type MoTe₂ transistors. (thermal release tape (TRT) and polymethyl methacrylate (PMMA)) **b** The obtained transistor arrays and integrated circuits based on 4-inch scale MoTe₂ thin films. (Scale bar 1 cm). **c** Optical image of the MoTe₂ transistor array. (Scale bar 500 μm). **d** Cross-sectional HAADF-STEM image of the fabricated MoTe₂ transistors. (Scale bar 20 nm). The inset illustration is a schematic diagram of a transistor structure. **e** Transfer characteristics of 100 sampling transistors (from the same die on the 4-inch wafer) for statistical analysis. (gate current $I_{gs}$.) **f–h** The histogram of the statistical distribution of the 100 devices: **f** mobility, **g** on/off ratio ($I_{on}/I_{off}$), and **h** threshold voltage ($V_{th}$).

there remains great potential for further dense integration through the reduction of device dimensions.

Figure 3c displays wafer-scale MoTe₂ transistor arrays with the same 14 μm × 100 μm channel dimensions (magnified view in Supplementary Fig. 13), while cross-sectional STEM imaging (Fig. 3d) confirms the structural integrity of the MoTe₂ transistors with sharp interfaces between the MoTe₂ channel and 16-nm HfO₂ dielectric. Electrical characterization of 100 individual sampled devices (Fig. 3e) demonstrates 100% operational yield with p-type enhancement-mode operation (complete current cut-off at $V_{gs}$ = 0 V). Here, enhancement-mode functionality enables single-supply operation, which is an advancement over conventional depletion-mode 2D transistors requiring dual voltage sources and larger power consumption. The superior characteristics of this high-performance field-effect transistor are attributable to two factors. First, we employed precursor engineering to achieve highly thickness-controlled growth of MoTe₂,

enabling the fabrication of uniform, ultrathin 3-layer channels across a 4-inch wafer. Second, the use of an ALD-grown HfO₂ dielectric with a high dielectric constant enables a substantially reduced equivalent oxide thickness (EOT). This increases gate capacitance and improves electrostatic control of the MoTe2 channel, resulting in a steeper subthreshold swing and a lower operating voltage. For a more detailed description of the uniformity of our fabricated MoTe₂ transistors, statistical analysis of the electrical performance parameters was conducted based on the transfer characteristics presented in Fig. 3e. The statistical results in Fig. 3f reveal a highly sharp distribution in carrier mobility of $7.04 \pm 1.65\ \mathrm{cm^2\ V^{-1}\ s^{-1}}$, offering good current driving capability necessary for further medium-scale IC construction. Here, the carrier mobility is evaluated via the formula $\mu = (L_c/W)(1/C_{ox})(1/V_{ds})(\partial I_{ds}/\partial V_{gs})$, where the oxide capacitance $C_{ox}$ of approximately 0.05 μF/cm² was measured by parallel-plate C-V measurements (details are provided in Supplementary Fig. 14).

In order to investigate the fundamental reasons for the relatively low carrier mobility in thin-film transistors (TFTs), the source/drain metal-semiconductor contact was analyzed using the Y-function method. As shown in Supplementary Fig. 15 and Supplementary Note 1, the contact resistivity of four classical MoTe$_2$ TFT is on the order of $10^8$ $\Omega\cdot\mu$m, suggesting the presence of poor metal-semiconductor interface quality. The contact resistance in our MoTe$_2$ TFT is relatively large compared with other reports, which reduces the source–drain current under the same applied voltage. As a result, the mobility extracted directly from the transfer characteristics is significantly under-estimated. We believe that this does not reflect an intrinsic limitation of the material itself; rather, the mobility could be further improved with optimized contact engineering that remains compatible with scalable fabrication processes. Figure 3g presents the statistical distribution of the on/off current ratio ($I_{on}/I_{off}$) for the p-type transistors, with an average value of $1.29 \times 10^5$ and a standard deviation of $3.68 \times 10^4$. This distribution ensures a clear distinction between the on and off states, enabling the transistors to act as effective switching devices. In Fig. 3h, the threshold voltage ($V_{th}$) shows a narrow distribution of $-3.38 \pm 0.042$ V. Here, $V_{th}$ is calculated by the intercept of the line extrapolated at the point of maximum transconductance (illustrated in Supplementary Fig. 16). The variation in $V_{th}$ is only 42 mV, accounting for only 1.07% of the 4 V operating gate voltage window, which is superior to previously reported p-type 2D transistors[18,19,21,26,27,34–40] and even approaching the foundry-fabricated silicon-based field-effect transistors at the 32 nm[43] and 45 nm[44] technology nodes, with standard deviations of 37 and 30 mV, respectively. Such high-uniformity in $I_{on}/I_{off}$ and $V_{th}$, greatly contribute to consistent switching characteristics across the operating voltage window, which lay a robust foundation for constructing ICs based on these transistors. It should be noted that, in order to validate the long-term stability of the fabricated MoTe$_2$ devices, they were subjected to extended storage under vacuum conditions. Transfer characteristics were measured after 14 months, as shown in the corresponding Supplementary Fig. 17 the devices exhibit negligible shift the transfer characteristics, also the extracted parameters (e.g., current on/off ratio (in Supplementary Fig. 17b) and threshold voltage (in Supplementary Fig. 17c)), confirming excellent operational stability over an extended long period. The devices were also exposed to ambient air, where performance degradation was observed, consistent with previous reports[18]. Although surface oxidation may induce p-type doping, adsorption of O$_2$ and H$_2$O at surface defects can introduce carrier scattering and moisture-related trapping that affect carrier transport. These results highlight the importance of effective encapsulation for maintaining device stability, which will be explored in future work. To evaluate device performance at the sub-micron scale, MoTe$_2$ transistors with varying channel lengths, including sub-micron dimensions, were characterized (Supplementary Fig. 18). As the channel length decreases, the devices retain well-behaved switching and stable transfer characteristics, with the on-state current increasing in accordance with typical scaling behavior. Although these devices with sub-micron dimensions exhibit slightly inferior uniformity compared to their counterparts with larger dimensions, their performance remains acceptable for circuit integration, demonstrating the practical utility of the grown MoTe$_2$ films.

## 2D PMOS logic: from uniform transistors to MSICs

The high uniformity of MoTe$_2$ transistors paves the way for the fabrication of logic gates and more complex integrated circuits. As a preliminary demonstration, logic gates that serve as the essential building block of digital ICs, including inverters (or the NOT gate), NAND, and NOR gates, are constructed using p-type metal-oxide-semiconductor (PMOS) logic. The inverter, depicted in Fig. 4a, employs two MoTe$_2$ transistors with distinct channel width-to-length ratios of 2 and 14, serving as the driver and active load, respectively (as illustrated in the

accompanying circuit diagram of Fig. 4b). Key electrical parameters of inverters that influence the cascading capability, such as the output logic level and noise margin (NM), can be derived from the voltage transfer characteristics (VTCs). The VTCs of a representative inverter, measured under varied biases of $V_{dd} = -2$ V, $-3$ V, and $-4$ V, are presented in Fig. 4c. Owing to the narrowly controlled distribution of threshold voltages and the substantial on/off ratio of the MoTe$_2$ transistors, the output voltage level can be effectively pulled up or down to $V_{dd}$ or GND (0 V), representing logical "1" and "0," respectively. As a result, near rail-to-rail output is achieved, which ensures good cascading capability for the integration of logic circuits. Figure 4d shows VTC (red solid line) and its mirror curve (blue dotted line) under a $V_{dd}$ bias of $-4$ V. The large area of the 'butterfly' diagram represents high noise tolerance with a noise margin of 63% for the inverter. These properties attribute great tolerance to the input noise, enabling the further development of more intricate ICs with increased logic depths. Leveraging the highly-uniform MoTe$_2$ transistors, fundamental PMOS logic NAND gate (Fig. 4e–g) and NOR gate (Fig. 4h–j) have been realized, demonstrating accurate time-varying Boolean output functionalities corresponding to logic "1" and "0," respectively. It is worth emphasizing that these gates exhibit nearly ideal high (5 V) and low (0 V) logic level outputs, thereby ensuring the cascading capability essential for further large-scale integration. Such results are attributed to the utilization of highly uniform MoTe$_2$ transistors with well-controlled threshold voltages in this study.

The construction of ring oscillators (ROs) by cascading the inverters is an effective and reliable way to evaluate the yield, uniformity, and speed of logic gates. Accordingly, ring oscillators with different stage numbers—5 stage (depicted in Supplementary Fig. 19), 15 stage (in Supplementary Fig. 20), and 59 stage (in Fig. 4k–m)—have been constructed utilizing the MoTe2-based inverters. At a supply voltage of 5 V, the oscillation frequencies achieved are 2065 Hz for the 5-stage RO, 816 Hz for the 15-stage RO, and 238 Hz for the 59-stage RO. Correspondingly, the propagation delays for single-state MoTe$_2$-based logic gates ($\tau = 1/2Nf$, where N is the RO stage number, f is the oscillation frequency) are 48.43 µs, 40.85 µs, and 35.61 µs, respectively. The consistency in propagation delay across ring oscillators with differing stage numbers strongly indicates the excellent performance uniformity and high yield of the basic logic gates. It is worth noting that the measured characteristics of the MoTe$_2$ transistors exhibit a small degree of hysteresis. As shown in Supplementary Fig. 21, the forward (red curve) and reverse (blue curve) transfer characteristics of a representative device display a slight shift. However, the magnitude of this hysteresis is very small compared to the operational voltage of 5 V, which would not affect practical circuit operation. Both the oscillation waveforms and logic outputs are stable, reproducible, and fully functional, underscoring the robustness of our device design and fabrication process. Future enhancements may be achieved by downsizing the MoTe2 transistors or increasing the supply voltage, thereby further reducing the gate propagation delay.

Given the robust logic operation of the basic gates demonstrated, attributed to the high transistor uniformity, a more complex arithmetic integrated circuit (IC) has been constructed using these MoTe$_2$ PMOS logic gates. The full adder is an important combinational logic circuit for arithmetic operations and is usually used as a basic building block for arithmetic logic units (ALUs). However, the implementation of a full adder based on p-type 2D transistors has been rarely reported due to the deep logic depth requirements, which necessitate a high level of uniformity and finely tuned output voltages for each constituent element. Here, as shown in Fig. 5a, we present a 4-bit full adder based on the PMOS MoTe$_2$ logic gates, which integrates 140 transistors and spans a logical depth of 12 stages. This is a medium-scale integrated circuit (MSIC, i.e., circuits with over 100 integrated devices) that has not previously been achieved using MoTe$_2$ or even other p-type 2D materials. The circuit design diagram of the 4-bit full adder is

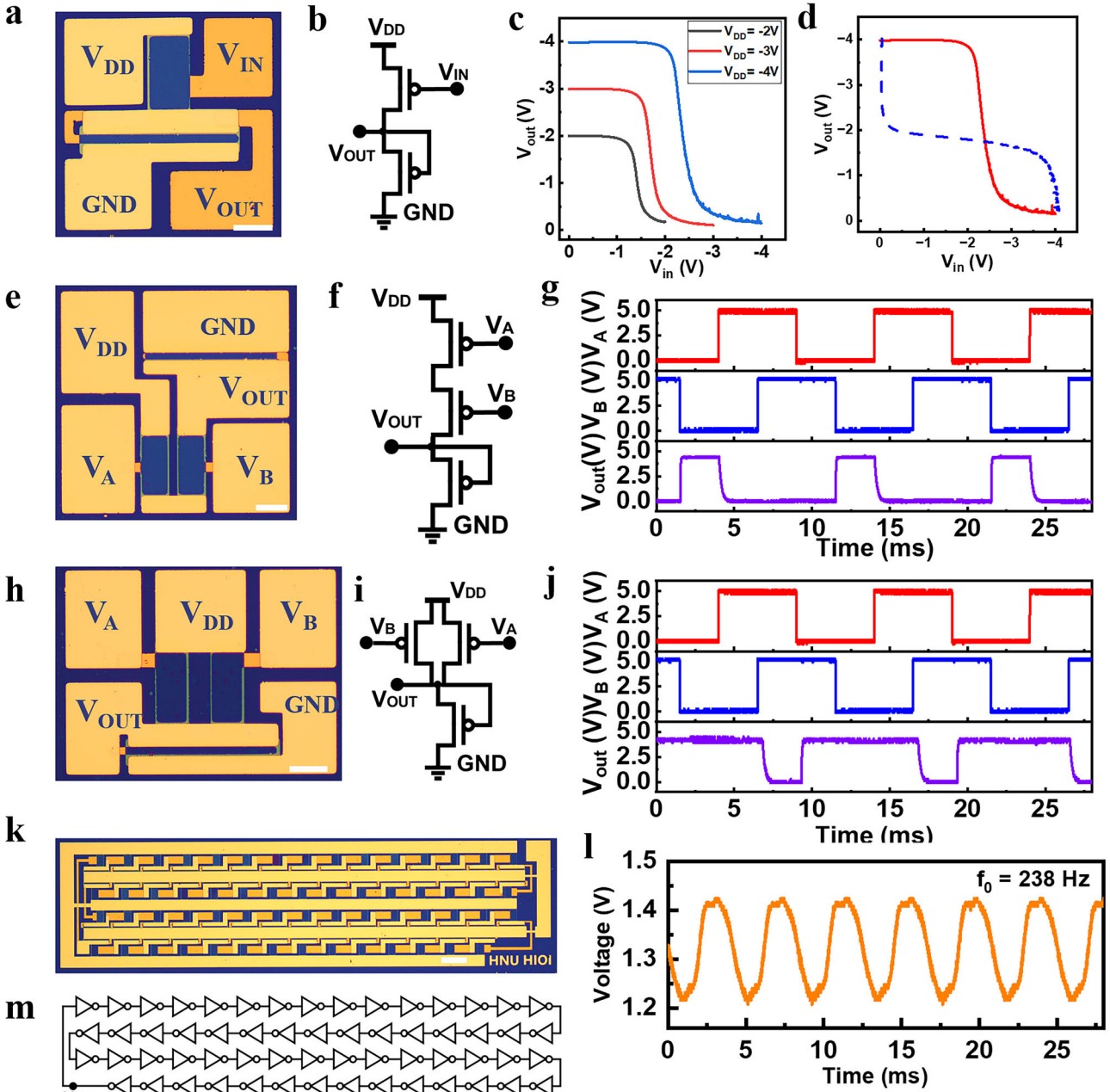

**Fig. 4 | Inverter, basic logic gates and ring oscillator characterization. a** The optical photograph of a PMOS logic MoTe$_2$-based inverter. (p-type metal-oxide-semiconductor (PMOS). output voltage ($V_{OUT}$), input voltage ($V_{IN}$), and supply voltage ($V_{DD}$)). (Scale bar 50 μm). **b** The circuit diagram of the p-type MoTe$_2$ inverter. **c** Voltage transfer curves (VTCs) of the inverter at different supplied $V_{DD}$ ranging from −2 V to −4 V. **d** VTC (solid red line) and its mirrored curve (dotted blue line) of the inverter under $V_{DD}$ of −4 V. **e-g** NOR gate. (input voltages ($V_A$ and $V_B$)). (Scale bar 50 μm). **h-j** NAND gate. (Scale bar 50 μm). **k–m** A 59-stage ring oscillator. (oscillation frequency ($f_0$)). (Scale bar 200 μm). Each panel above includes the optical image, circuit diagrams, and input-output characteristics.

portrayed in Fig. 5b, which incorporates four sequential 1-bit full adders. Each sub-component, the 1-bit full adder, is composed of three NAND gates and two XOR gates with 35 p-type MoTe$_2$ transistors (illustrated in Fig. 5c). Here, the input terminals A and B in Fig. 5b, c represent the inputs to be summed, with C indicating the carry-in from previous additions. The output terminal S denotes the sum, while C represents the carry-out for subsequent arithmetic operations.

To verify the operational accuracy of the 4-bit MoTe$_2$ full-adder, we conducted a comprehensive test of the output logic values across 16 distinct addition combinations for the input set ($A_3 A_2 A_1 A_0$, $B_3 B_2 B_1 B_0$), specifically including pairs such as (1110, 1111), (0101, 1101), (0011, 1100), and so forth, as detailed in Fig. 5d. The results confirm the correct logical functionality of the 4-bit full adder., illustrating that the

4-bit full-adder exhibits right time-varying Boolean logical functions. As depicted in Fig. 5e, a comparison is implemented of wafer size and integrated device counts with previous works based on p-type 2D materials[14,15,18–21,23,26,27,34–40,45–60]. From the comparison, it can be concluded that although there have been prior instances of arrayed transistors[14,15,18,19,34,35,38,39] or simple logic gate circuits[20,23,48,51,54,56–59] fabricated from p-type 2D materials, the realization of the medium-scale integrated circuit, such as the 4-bit adder demonstrated here, marks a significant advancement to meet the criteria for practical application in terms of scale and wafer size. In contrast[19], our approach utilizes nearly standard nanodevice fabrication processes and conventional contact materials, enabling highly uniform and reproducible device performance across a 4-inch wafer. We achieve an on/off ratio of

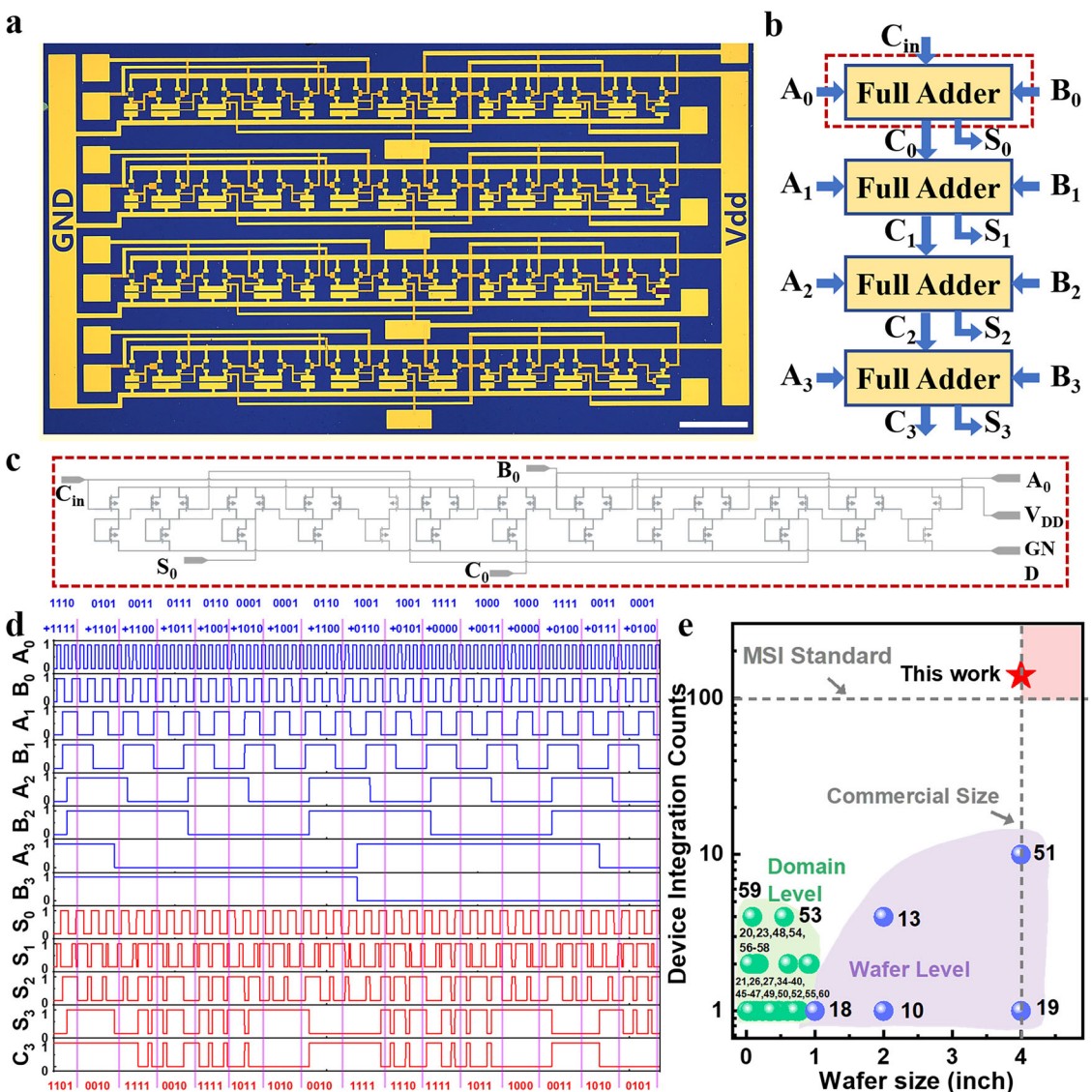

**Fig. 5 | Characterization of the MoTe₂ -based 4-bit full adder. a** Optical image of the 4-bit full adder. (Scale bar 500 µm). **b** System design of the 4-bit full adder. (The input terminals $A_{0-3}$ and $B_{0-3}$ represent the inputs to be summed, with $C_{in}$ and $C_{0-2}$ indicating the carry-in from previous additions. The output terminal $S_{0-3}$ and $C_3$ denote the sum.) **c** Circuit diagram of a 1-bit full adder. **d** Input-output characterization of the 4-bit full adder with the input of 16 kinds of addition combinations. **e** A comparison of wafer size and integrated device counts between this work and previous reports based on p-type 2D materials. (dashed line represents MSI Standard and Commercial Size)[14,15,18–21,23,26,27,34–40,45–60].

~$1.47 \times 10^5$—comparable and even superior to the reported values—while simultaneously demonstrating a functional medium-scale integrated circuit comprising over 100 p-type 2D transistors. This balance between high performance and process scalability represents a key step toward practical applications of p-type 2D semiconductors in integrated electronics. Furthermore, as shown in Supplementary Table 1, this successful demonstration showcases a substantial increase of p-type 2D circuits in a deeper logical depth and a denser integration density. These achievements signify that integrated circuits utilizing p-type 2D transistors are progressing towards the medium-scale domain, which would expand the horizons for their application in future advanced electronic systems.

## Discussion

In conclusion, we have successfully overcome many of the current limitations in the large-scale integration of p-type 2D semiconductors by scaling from reported discrete device demonstrations to medium-scale integrated circuits (MSICs) on industry-compatible 4-inch wafer substrates. Through a synergistic optimization of material growth, device engineering, and fabrication processes, we have demonstrated the potential of p-type 2D semiconductors for large-scale electronic applications. The high uniformity and performance of the synthesized 2H-MoTe₂ films, coupled with the good performance of the resulting transistors, including large current on-off ratios, underscore the viability of our approach. It is worth noting that the device dimensions in this work are still relatively large (although comparable to those reported for state-of-the-art p-type 2D circuits). Sub-micron (<1 µm) devices are critical, and we plan to pursue further device scaling in future work to achieve higher performance. Moreover, the fabrication flow still relies on film transfer to ensure device performance and process yield. Owing to the absence of dangling bonds on two-dimensional materials' surfaces, conventional atomic layer deposition is nucleation-limited and struggles to form uniform, conformal, high-quality gate dielectrics, thereby hindering direct top-gate integration and necessitating the transfer of 2D materials in this work. The development of transfer-free, industry-compatible, high-quality

dielectric deposition and in situ integration strategies is expected to simplify the fabrication process and facilitate the scalable implementation of two-dimensional CMOS technologies.

## Methods

### Growth of ultrathin 2H-MoTe$_2$ Films

The MoTe$_2$ films were synthesized by tellurizing the Mo films at atmospheric pressure in a horizontal tube furnace equipped with mass flow controllers and a vacuum pump. First, the substrate was treated with high-energy oxygen plasma (inductively coupled in a CF$_4$/O$_2$ atmosphere) to alter the substrate surface hydrophilicity. Subsequently, we employed magnetron sputtering with molybdenum as the target material to deposit a molybdenum film. To ensure uniform deposition on 4-inch wafers, we controlled the deposition rate between 0.1–0.3 Å s$^{-1}$ by adjusting the sputtering DC power (8 W) and chamber pressure (0.7–0.8 Pa). Then, the tellurium powder (99.9%, Alfa Aesar) is put into the alumina boat, and the alumina boat is inserted into a 5-inch diameter quartz tube in the furnace. The growth temperature was set at 450 °C, the Ar flow rate was 80 sccm, and the growth time was 10 minutes. The Mo source is placed vertically in the middle of the quartz tube; the Te source and the molecular sieve are placed together in the alumina boat, which is then inserted into the upstream position of the 5-inch-diameter quartz tube in the furnace. When the quartz tube is vacuumized to below 10 mTorr, high-purity Ar gas is introduced, and the flow rate is 500 standard cubic centimeters per minute (sccm) until the atmospheric pressure is reached. This process was repeated three times. After that, the Ar flow rate was set to 60 sccm, and high-purity H$_2$ gas was introduced at 80 sccm. The growth temperature and time were 600 °C and 2 h, respectively. After the growth period, the furnace was turned off and naturally cooled to room temperature.

### Transfer of MoTe$_2$ films

First, a layer of PMMA is spin-coated on a MoTe$_2$/Si/SiO$_2$ substrate at 4000 RPM and then baked at 150 °C for 5 minutes. Then, the hot release tape is applied to the sample and the sample is immersed in 10% KOH solution for 30 minutes. Peel the hot strip tape /PMMA /MoTe$_2$ layer with tweezers, wash it several times in pure water, and then transfer the layer to the target substrate. Heat thermal release tapes (TRT) are used to keep the MoTe$_2$ film flat and release the PMMA/MoTe$_2$ layer onto the substrate at 120 °C. After complete release, the sample was washed in acetone for 5 minutes to remove all polymers.

### MoTe$_2$ transistor fabrication and electrical characterization

For back-gated MoTe$_2$ field-effect transistors, first, the Ti/Au (5/20 nm) local gate is defined by standard lithography on the substrate, followed by electron-beam evaporation (EBE) and lift-off. Secondly, atomic layer deposition was employed to prepare 16 nm HfO$_2$ or 20 nm Al$_2$O$_3$ insulating layers. The HfO$_2$ film is deposited at a deposition temperature of 250 °C using tetrakis (ethylmethylamido) hafnium and H$_2$O as the precursors. The tetrakis (ethylmethylamido) hafnium precursor was pulsed for 500 ms with a purge time of 5 s. The H$_2$O precursor was pulsed for 500 ms with a purge time of 8 s. The Al$_2$O$_3$ film is deposited at 250 °C using trimethylaluminium and H$_2$O as the precursors. The trimethylaluminium precursor was pulsed for 500 ms with a purge time of 5 s. The H$_2$O precursor was pulsed for 500 ms with a purge time of 8 s. Subsequently, photolithography was performed using standard photolithography techniques, followed by process etching employing inductively coupled plasma reactive ion etching (ICP-RIE) under an atmosphere of 80 W RF bias power with SF$_6$/O$_2$/Ar = 120/40/40 s. c. c. m. The MoTe$_2$ film is then transferred to the substrate, and the source and drain on the MoTe$_2$ film of the device are prepared using lithography and electron beam evaporation of Ti/Pd/Au (1/40/20 nm). Finally, the MoTe$_2$ films are patterned using photolithography and the

aforementioned ICP-RIE etching process. All electrical measurements were taken under moderate vacuum conditions. using a semiconductor parameter analyzer (Keysight B1500), digital oscilloscope (Siglent SDS1024X), and signal generators (Siglent SDG1022X). The output analysis of the 4-bit full adder was implemented by a logic analyzer (DSLogic, DreamSourceLab, Co., Ltd.). Better electrical performance is expected if the vacuum level in the metal evaporation chamber can be improved.

### Characterization of the morphology and structure

Raman spectra of the CVD-grown few-layer 2H-MoTe$_2$ Films were collected using a confocal Raman microscope system (Alpha300, WiTec) with an excitation wavelength of 532 nm. The Raman spectra were obtained at a laser power of 200 μW and an accumulation time of 10 s (using a grating of 1800 grooves per mm). The atomic structure and chemical composition of the MoTe$_2$ were investigated by STEM, energy dispersive X-ray spectroscopy (EDS), high-resolution STEM (HRSTEM), and selected-area electron diffraction (SAED). X-ray photoelectron spectroscopy (XPS, SHIMADZU AXIS SUPRA+) was used to obtain the MoTe$_2$ signals.

## Data availability

The data of this report has been included in the published article and its Supplementary Information. Additional raw data are available from the corresponding authors upon request.

## Code availability

All the codes that support the findings of this study are available from the corresponding authors upon request.

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

## Acknowledgements

This work was supported in part by the National Key R&D Program of China under Grant 2024YFB4405300 (L.X.), 2022YFA1402501 (A.P.) and 2022YFA1204300 (A.P.); the National Natural Science Foundation of China under Grant 62101181 (L.X.),52221001 (A.P.), 62090035 (A.P.), 52372146 (A.P.) and U22A20138 (A.P.); the Natural Science Foundation of Hunan Province under Grant 2023JJ20016 (L.X.); and the Key Research and Development Plan of Hunan Province under Grant 2022GK3002 (A.P.) and 2023GK2012 (A.P.); and the Hunan Provincial Innovation Foundation For Postgraduate CX20240403 (H.W.).

## Author contributions

L.X. conceived the experiment. L.X. and A.P. supervised this project. L.X. and H.W. designed the devices and circuits. H.W. fabricated the devices, performed the electrical measurements, and analyzed and interpreted the data with input from Z.L., B.Z., Z.T., H.Y., Y.Y., Y.W., H.Z., Q.S., H.L, G.W., D.L., and H.W. prepared the MoTe2 thin films. The manuscript was written with contributions from all authors, and all authors approved the final version of the manuscript.

## Competing interests

The authors declare no competing interests.
