## [Transparent Peer Review file · Nature Communications]

Medium-scale integrated circuits based on p-type 2D semiconducting MoTe₂

Corresponding Author: Professor Anlian Pan

Version 1:

Reviewer comments:

Reviewer #1

(Remarks to the Author)

The authors have adequately addressed the previous concerns regarding the stability of the MoTe₂ devices and have improved the overall quality of the manuscript. Therefore, this reviewer recommend it for publication.

Reviewer #2

(Remarks to the Author)

In the revised version of their manuscript the authors provide additional data to support the original claims of their publication, that their uniform growth of 2H-MoTe₂ at a wafer scale enables the fabrication of highly uniform p-type transistors that can be used to produce medium scale integrated logic circuits with industry compatible techniques. The additional data in the SI on the characterisation of their grown thin films provides insight as to their quality. The scale up from previously reported 39 transistor p-type circuits to a 4 bit full adder medium scale integrated circuit is an advancement over previous publications for p-type 2D material transistors.

This work presents an alternative approach to other p-type 2D material growth, and the revised manuscript clarifies that the advance is the scale up to 4 inch wafers of 3 layer 2H-MoTe₂, rather than the growth methods themselves, the films produced in this work are polycrystalline with lots of individual grains and a relatively rough surface. The authors claim that their growth method is comparable with existing industry techniques for applications however the MoTe₂ film's novel uniformity is uncertain on a scale useful for technology.

Whilst the results presented here are interesting and worth reporting I do not believe their novelty is sufficient for publication in nature communications and that they would be suited to a more specialised publication.

I provide the following more detailed comments to the authors.

1. The quality of the 2H-MoTe₂ films

The authors have conducted significant extra work to characterise their thin MoTe₂ films in more detail compared to the original manuscript showing AFM characterisation and TEM surface characterisation. Whilst the new characterisation does indicate consistency across the wafer in the new fig S9, on a smaller scale it does re-enforce my original concerns about the quality of the grown MoTe₂. The AFM analysis in figure S8 shows the grain morphology with many connected grains. Additionally the averaged roughness from fig S9 is high, 0.3nm average roughness with individual high points of several nm. Compared with a layer thickness for MoTe₂ of 0.6-0.65nm, this would be challenging to work with for further device development. The surface quality would be an additional hinderance for inclusion of the grown MoTe₂ into more advanced heterostructures, either for atmospheric protection or, as mentioned by my fellow reviewer, for top gated devices.

2. The environmental stability of MoTe₂

In the revised manuscript the authors confirm that their devices are stable over a period of 14 months with minimal change in transfer characteristics whilst under vacuum. Although this is sufficient for this demonstration, as the paper is heavily

focussed of the applicability of the growth technique for industry over other published methods and its comparability with existing processes, it is important that devices can continue to perform after storage in ambient conditions as well as only under vacuum storage.

3. Device size.

The authors acknowledge the importance of sub micron devices in the revised manuscript and highlight the importance of these devices for future work as the large device size presented of $14 \times 100 \mu\text{m}$ will average out imperfections due to grain boundaries and defects present in the grown films. The devices presented are of a similar size to those shown in previously published work on p-type 2D material transistors. Smaller devices would likely have a much less uniform distribution of key transport characteristics. Whilst uniformity in smaller devices is not required for the key aim of this paper, to demonstrate for the first time a medium scale integrated circuit with p-type MoTe₂, this miniaturisation would show the usefulness of the growth techniques to modern technological applications.

Reviewer #3

(Remarks to the Author)

The manuscript by Wang et al. presents a significant milestone in the field of 2D electronics. The authors report the fabrication of the first medium-scale integrated circuit (MSIC) based on p-type 2D semiconductors, specifically realizing a 140-transistor full adder. This achievement is underpinned by their successful synthesis of highly uniform, wafer-scale (4-inch) 2H-MoTe₂ films via a clever precursor engineering strategy.

The work addresses a critical bottleneck in the field: while n-type 2D devices have seen rapid progress, scalable and high-performance p-type counterparts have lagged behind. The reported electrical uniformity (e.g., small threshold voltage variation) and the successful demonstration of complex logic gates represent a substantial leap from discrete device demonstrations toward practical integrated circuits. The experimental design is rigorous, and the results are convincing.

I recommend the publication of this manuscript. However, I have listed a few specific queries and suggestions below regarding the material growth mechanism and fabrication processes. Addressing these points will further strengthen the manuscript and provide valuable insights to the community.

Specific Comments:

Growth Mechanism and Nucleation: The authors synthesize the MoTe₂ films by tellurizing Mo precursor films. Could the authors clarify the phase evolution during this process? specifically, does the reaction lead directly to the formation of the 2H phase, or does it involve a phase transition (e.g., from a metastable 1T' phase to 2H phase)? Furthermore, the work highlights the use of ultrathin Mo precursors. It would be scientifically valuable to discuss how the thickness of the precursor, especially in the ultra-thin regime (limiting case), influences the nucleation density and growth mode of the MoTe₂.

Crystalline Domain Size: The manuscript provides excellent data on the macroscopic uniformity of the film across the 4-inch wafer. However, as the film is polycrystalline, quantitative information regarding the microstructure is necessary. Please provide data or an estimation of the average crystalline domain size (grain size) of the synthesized thin films.

Experimental Parameters: To ensure the reproducibility of this impressive work, the "Methods" section requires more specific details. Please explicitly list the detailed parameters for the micro-fabrication steps, specifically for the Reactive Ion Etching (RIE), Inductively Coupled Plasma (ICP) etching, and magnetron sputtering processes (e.g., power, pressure, gas flow rates, and bias voltage where applicable).

Device Integration and Interface Engineering:

Transfer Process: The current fabrication flow relies on transferring the MoTe₂ films. While successful for this demonstration, transfer processes inevitably introduce defects, interface impurities, and yield challenges that are generally incompatible with large-scale CMOS manufacturing. It would be beneficial for the authors to acknowledge these limitations and briefly discuss pathways toward transfer-free (direct growth) integration in the future.

Dielectric Integration: The authors utilize a local back-gate configuration, likely to preserve the channel quality. Top-gate integration is typically preferred for high-density scaling but is often hindered by damage or unintended doping during the Atomic Layer Deposition (ALD) of high-k dielectrics. I suggest the authors discuss potential strategies to solve or mitigate the ALD-induced doping/damage on the MoTe₂ channel, which would be a necessary step for future top-gate MSIC implementation.

Reviewer #4

(Remarks to the Author)

This manuscript reports the growth of 4-inch, wafer-scale 2D MoTe₂ thin films via the optimization of the tellurium source supply. Using these as-prepared thin films, the authors successfully demonstrated the fabrication of a 140-transistor full adder, which illustrates the potential of their material for 2D CMOS circuit integration. I recommend accepting this manuscript, provided that the authors appropriately address the following comments.

1.Regarding the novelty of the growth method: The manuscript highlights a precursor engineering strategy. However, the authors should more clearly delineate the fundamental advancements of their method compared to the early CVD growth of 2H-MoTe₂ reported a decade ago (J. Am. Chem. Soc. 2015, 137, 37, 11892–11895) and subsequent studies. The current description suggests modifications rather than a paradigm shift. A more detailed discussion of the specific innovations and their impact would be beneficial.

2.Regarding the evidence for uniformity and quality: The claim of high quality and uniformity across the 4-inch wafer is central to this work. To more robustly substantiate this claim, the authors should provide structural characterization, such as selected area electron diffraction (SAED) patterns, from multiple, well-defined locations across the wafer. This data would serve as crucial evidence to support the asserted uniformity.

3.Regarding the device performance: The authors report a field-effect mobility of ~7 cm²/Vs. This value is lower than what has been previously achieved for MoTe₂ FETs (often in the range of 10-50 cm²/Vs), which creates a potential conflict with the claim of “high-quality” material. Furthermore, the recent report of 2D β-Bi₂O₃ with a hole mobility of 136.6 cm²/Vs (Nat. Mater. 24, 688–697, 2025) sets a new benchmark for p-type 2D semiconductors. To enhance the potential of the 2H MoTe₂ synthesized by authors, it is critical to improve the mobility of single FET before pushing to integrated devices.

Reviewer #5

(Remarks to the Author)

Version 2:

Reviewer comments:

Reviewer #2

(Remarks to the Author)

The authors have included further characterisation of their MoTe₂ devices. The additional SAED measurements of the wafer scale grown MoTe₂, discussion of the MoTe₂ domain size, tests to explore the effects of ambient storage on these devices, and initial scalability experiments to produce smaller devices strengthen the author’s claims and improve the overall quality of the manuscript. With these new measurements, and their related discussion in the text the authors have addressed my previous concerns and I recommend this paper for publication in Nature Communications.

Reviewer #3

(Remarks to the Author)

I would like to thank the authors for their hard work and the comprehensive point-by-point response to the reviewers' comments. The revised manuscript has been substantially improved. Before final acceptance, I suggest the following minor refinements: The author observes a decrease in the ON-state current (I_{on}) after exposing the device to air. However, it is widely reported in the literature that the surface oxidation of MoTe₂ typically results in the formation of MoO_x, which induces a strong p-doping effect. Theoretically, this p-doping should lead to an increase in the I_{on} for p-type or ambipolar MoTe₂ devices. The author should provide a detailed explanation or additional experimental evidence to clarify why the current decreased in this specific case, addressing whether factors such as increased scattering, contact resistance degradation, or moisture-induced trapping are dominant.

Reviewer #4

(Remarks to the Author)

The authors have adequately addressed my previous concerns. Therefore, I recommend accepting this manuscript in its current form.

Reviewer #5

(Remarks to the Author)

Response to the Reviewers' comments:

Response to Reviewer #1

The authors have adequately addressed the previous concerns regarding the stability of the MoTe₂ devices and have improved the overall quality of the manuscript. Therefore, this reviewer recommend it for publication.

Response: We sincerely thank the reviewer for their positive assessment of our work and for recognizing the improvements made regarding the stability of MoTe₂ devices. We greatly appreciate the reviewer's evaluation of the manuscript and their recommendation for publication, which is invaluable to us.

Response to Reviewer #2

1. The authors have conducted significant extra work to characterise their thin MoTe₂ films in more detail compared to the original manuscript showing AFM characterisation and TEM surface characterisation. Whilst the new characterisation does indicate consistency across the wafer in the new fig S9, on a smaller scale it does re-enforce my original concerns about the quality of the grown MoTe₂. The AFM analysis in figure S8 shows the grain morphology with many connected grains. Additionally, the averaged roughness from fig S9 is high, 0.3nm average roughness with individual high points of several nm. Compared with a layer thickness for MoTe₂ of 0.6-0.65nm, this would be challenging to work with for further device development. The surface quality would be an additional hinderance for inclusion of the grown MoTe₂ into more advanced heterostructures, either for atmospheric protection or, as mentioned by my fellow reviewer, for top gated devices.

Response: We sincerely thank the reviewer for this insightful comment. Here, it should be emphasized that, in order to rigorously evaluate the morphological quality of the MoTe₂ films, decoupling the intrinsic roughness of the films from the baseline topography of the silicon substrate is essential. As illustrated in Fig. R1, the standard Si/SiO₂ substrate exhibits an average roughness of 0.2 nm. As the as-grown MoTe₂ film has an average roughness of 0.3 nm, the net contribution of the surface roughness from the MoTe₂ growth process is merely ~0.1 nm. This marginal increase suggests that the film maintains atomic-level planarity and good conformality to the substrate surface. Crucially, the achieved roughness of 0.3 nm is comparable to state-of-the-art

benchmarks and has been shown to be sufficiently robust for subsequent device integration [shown in ref. 18[Ra~0.25 nm], 19[Ra<0.90 nm] of the manuscript].

Figure R1. a. Detailed atomic force microscopy surface topography images of 6 sampling points on Si/SiO₂. Scale bar 1 μm. **b.** Atomic force microscopy surface topography images of Si/SiO₂ at the 1 μm scale. Scale bar 200 nm. **c.** Atomic force microscopy surface topography images of Si/SiO₂ at the 5 μm scale. Scale bar 1 μm. **d.** Atomic force microscopy surface topography images of Si/SiO₂ at the 15 μm scale. Scale bar 3 μm.

2. In the revised manuscript the authors confirm that their devices are stable over a period of 14 months with minimal change in transfer characteristics whilst under vacuum. Although this is sufficient for this demonstration, as the paper is heavily focussed of the applicability of the growth technique for industry over other published methods and its comparability with existing processes, it is important that devices can continue to perform after storage in ambient conditions as well as only under vacuum storage.

Response: We thank the reviewer for raising the important issue of the environmental stability of MoTe₂ devices. Following the reviewer's suggestion, we performed periodic electrical measurements on devices stored under ambient air conditions. Over a 7-day exposure period (Fig. R2), the devices exhibited some degree of performance degradation, primarily manifested as small threshold-voltage shifts and a slight reduction in on-state current. We attribute these changes mainly to the absence of effective device encapsulation at the present stage, which allows adsorption of ambient species such as H₂O and O₂ at the MoTe₂ surface or interfaces. Such adsorption can introduce non-intrinsic defects, thereby affecting carrier transport.

Importantly, despite these performance variations, the fundamental transistor functionality of the devices remains intact. Throughout the ambient exposure period, the devices continue to operate as normally switching field-effect transistors, and the current on/off ratio remains at a level sufficient to support logic-state switching in digital circuits. This indicates that the devices could retain functional stability for certain periods.

In contrast, our results clearly demonstrate that the devices exhibit excellent stability under vacuum storage, maintaining nearly unchanged electrical characteristics over a period of 14 months. Based on this observation, we believe that the introduction of appropriate device-level or wafer-level encapsulation strategies would effectively suppress environmental adsorption effects and enable similarly robust stability under ambient conditions.

Finally, we would like to emphasize that the primary objective of this work is not to establish the ultimate limits of device stability, but rather to demonstrate the feasibility of realizing medium-scale integrated circuits based on wafer-scale p-type two-dimensional semiconductors. The manuscript is therefore focused on validating that p-type 2D materials can support complex digital circuit functionality at the integration level. We appreciate the reviewer's valuable suggestion regarding stability, and investigations into long-term ambient stability and encapsulation strategies will constitute an important focus of our future work toward practical p-type two-dimensional integrated circuits.

Figure R2. Transfer characteristics of the FET measured during a 7-day exposure period in an air atmosphere.

3. The authors acknowledge the importance of sub micron devices in the revised manuscript and highlight the importance of these devices for future work as the large device size presented of $14 \times 100 \mu\text{m}$ will average out imperfections due to grain boundaries and defects present in the grown films. The devices presented are of a similar size to those shown in previously published work on p-type 2D material transistors. Smaller devices would likely have a much less uniform distribution of key transport characteristics. Whilst uniformity in smaller devices is not required for the key aim of this paper, to demonstrate for the first time a medium scale integrated circuit with p-type MoTe_2 , this miniaturisation would show the usefulness of the growth techniques to modern technological applications.

Response: We thank the reviewer for emphasizing the importance of sub-micron devices in assessing material uniformity and technological relevance. In response, we included additional measurements on devices with varying channel lengths (shown in Fig. S18 of the revised supporting information).

As shown in Fig. S18b, with the down-scaling of channel length, the MoTe₂ based transistors continue to exhibit well-behaved switching functionality and normal transfer characteristics. The on-state current increases, which is consistent with typical down-scaling behaviors. Meanwhile, we performed statistical measurements on multiple devices with sub-micron dimensions (Fig. S18d). Despite the expected short-channel effects at reduced sizes, the devices still maintain good uniformity with limited device-to-device variation in transfer characteristics. Although these devices with sub-micron dimensions exhibit slightly inferior uniformity compared to their counterparts with larger dimensions, their performance remains acceptable for circuit integration, demonstrating the practical utility of the grown MoTe₂ films.

Finally, we note that the primary goal of this work is to demonstrate medium-scale integrated circuits based on wafer-scale p-type two-dimensional materials. Systematic sub-micron scaling studies will be pursued in future work. The corresponding data and discussion have been added to the revised manuscript.

In page 20 of the revised Supporting information:

Figure S18. MoTe₂ transistors with different channel dimensions. a, Optical micrograph of MoTe₂ field-effect transistors with varying channel lengths (L). Scale bar 9 μm . b, Transfer characteristics of MoTe₂ transistors with different channel lengths measured under identical conditions. c, Optical image of an array of MoTe₂ transistors with scaled device dimensions. Scale bar 90 μm . d, Transfer characteristics of 12 representative devices with identical geometry, showing good device-to-device uniformity.

In page 12 of the revised manuscript:”

“To evaluate device performance at the sub-micron scale, MoTe₂ transistors with varying channel lengths, including sub-micron dimensions, were characterized (in Fig. S18). As the channel length decreases, the devices retain well-behaved switching and stable transfer characteristics, with the on-state current increasing in accordance with typical scaling behavior. Although these devices with sub-micron dimensions exhibit

slightly inferior uniformity compared to their counterparts with larger dimensions, their performance remains acceptable for circuit integration, demonstrating the practical utility of the grown MoTe₂ films.”

Response to Reviewer #3

The manuscript by Wang et al. presents a significant milestone in the field of 2D electronics. The authors report the fabrication of the first medium-scale integrated circuit (MSIC) based on p-type 2D semiconductors, specifically realizing a 140-transistor full adder. This achievement is underpinned by their successful synthesis of highly uniform, wafer-scale (4-inch) 2H-MoTe₂ films via a clever precursor engineering strategy.

The work addresses a critical bottleneck in the field: while n-type 2D devices have seen rapid progress, scalable and high-performance p-type counterparts have lagged behind. The reported electrical uniformity (e.g., small threshold voltage variation) and the successful demonstration of complex logic gates represent a substantial leap from discrete device demonstrations toward practical integrated circuits. The experimental design is rigorous, and the results are convincing.

I recommend the publication of this manuscript. However, I have listed a few specific queries and suggestions below regarding the material growth mechanism and fabrication processes. Addressing these points will further strengthen the manuscript and provide valuable insights to the community.

Response: We sincerely thank the reviewer for the assessment of our work and for the encouraging comments regarding the clarity and quality of the manuscript. We have carefully considered all the comments provided by the referee, and have revised the manuscript accordingly to further improve its quality, which are present point-by-point as follows.

1. Growth Mechanism and Nucleation: The authors synthesize the MoTe₂ films by tellurizing Mo precursor films. Could the authors clarify the phase evolution during this process? Specifically, does the reaction lead directly to the formation of the 2H phase, or does it involve a phase transition (e.g., from a metastable 1T' phase to 2H phase)? Furthermore, the work highlights the use of ultrathin Mo precursors. It would be scientifically valuable to discuss how the thickness of the precursor, especially in the ultra-thin regime (limiting case), influences the nucleation density and growth mode of

the MoTe₂.

Response: We thank the reviewer for the insightful comments regarding the growth mechanism and phase evolution during the tellurization of Mo precursor films.

Based on our systematic experimental observations during growth optimization, which are consistent with previously reported phase-transition behaviors [such as ref. 17, 18, 26 in the manuscript], the formation of MoTe₂ in this work does not proceed via direct crystallization into the 2H phase. Instead, we identify a clear phase evolution process. At the early stage of tellurization, when Te availability is limited, the metastable 1T'-MoTe₂ phase preferentially forms. As the tellurization process continues with sustained Te supply, a gradual transformation from the 1T' phase to the thermodynamically more stable 2H phase occurs. Importantly, during growth we observe the coexistence of 1T' and 2H phases, as shown in Fig. S4, which provides direct experimental evidence that the final 2H phase is achieved through a phase transition rather than by direct formation.

In our growth process, the thickness of the Mo precursor plays a decisive role in determining the final thickness of the converted MoTe₂ films, which have been experimentally demonstrated in Fig. 2c of the manuscript. Specifically, the Mo precursor thickness directly defines the total amount of Mo available for conversion during tellurization, thereby imposing an intrinsic upper limit on the achievable MoTe₂ layer number. By reducing the Mo precursor into the ultrathin regime, the growth process enters a Mo-limited conversion regime, in which both nucleation density and vertical growth are constrained by the finite Mo supply. Under these conditions, excessive multilayer stacking is suppressed, and the growth proceeds toward few-layer MoTe₂ with well-defined thickness. This leads to a reduced and more controllable nucleation density and favors a continuous, thickness-limited growth mode rather than uncontrolled multilayer accumulation. As a result, precise control over the Mo precursor thickness enables deterministic regulation of the final MoTe₂ thickness and promotes the formation of uniform few-layer films over large areas.

To provide a visual representation of the material growth process, we have added the corresponding images and descriptions in the revised manuscript and supporting information, which are present as below.

In page 6 of the revised Supporting information:

Figure S4. Optical image showing the coexistence of 1T' and 2H phases of MoTe₂. Scale bar 200 μm.

In page 6 of the revised manuscript:

“To elucidate the growth mechanism during tellurization, we find that the formation of MoTe₂ does not proceed via direct crystallization into the 2H phase. Instead, a metastable 1T' phase forms at the early stage and subsequently transforms into the thermodynamically stable 2H phase with continued Te supply, with the coexistence of both phases observed during growth providing direct experimental evidence for this phase-transition pathway (Fig. S4).”

In page 8 of the revised manuscript:

“In addition, the thickness of the Mo precursor defines the total amount of Mo available for conversion, thereby imposing an intrinsic constraint on the final MoTe₂ layer number (Fig. 2c). When the precursor approaches the ultrathin regime, the growth enters a Mo-limited conversion mode that suppresses excessive nucleation and vertical stacking, enabling the formation of thickness-controlled and uniform few-layer films.”

2. Crystalline Domain Size: The manuscript provides excellent data on the macroscopic uniformity of the film across the 4-inch wafer. However, as the film is polycrystalline, quantitative information regarding the microstructure is necessary. Please provide data or an estimation of the average crystalline domain size (grain size) of the synthesized thin films.

Response: We thank the reviewer for emphasizing the importance of providing quantitative information on the microstructure of the polycrystalline MoTe₂ films. Although the MoTe₂ films synthesized in this work are polycrystalline, their grain sizes are relatively large. During the growth process, we clearly observe an intermediate state characterized by the coexistence of the 1T' and 2H phases, as shown in Fig. S4. These phase-coexisting regions provide direct real-space evidence of laterally continuous crystalline domains and allow an estimation of the characteristic grain domain size. As inferred from Fig. S4, the largest continuous crystalline domain reaches approximately 900 μm, approaching the millimeter scale. Moreover, even in regions where the film has fully merged into a continuous layer, the typical crystalline domain size remains on the order of hundreds of micrometers. This behavior indicates a low nucleation density and a growth mode dominated by lateral domain expansion. Consequently, despite being polycrystalline, the film's large crystalline domains result in a substantially reduced density of grain boundaries, with typically only one or two intersecting a transistor channel. This microstructural feature is a critical factor enabling uniform device performance and the successful fabrication of medium-scale integrated circuits based on MoTe₂.

In page 6 of the revised manuscript:

“As inferred from Fig. S4, the largest continuous crystalline domain reaches approximately 900 μm, approaching the millimeter scale. Consequently, despite being polycrystalline, the film's large crystalline domains result in a substantially reduced density of grain boundaries, with typically only one or two intersecting a transistor channel. This microstructural feature is a critical factor enabling uniform device performance and the successful fabrication of medium-scale integrated circuits based on MoTe₂.”

3.Experimental Parameters: To ensure the reproducibility of this impressive work, the "Methods" section requires more specific details. Please explicitly list the detailed parameters for the micro-fabrication steps, specifically for the Reactive Ion Etching (RIE), Inductively Coupled Plasma (ICP) etching, and magnetron sputtering processes (e.g., power, pressure, gas flow rates, and bias voltage where applicable).

Response: We sincerely thank the reviewer for highlighting the importance of experimental parameters. We fully agree that providing precise fabrication parameters

is essential for the community to replicate and build upon our work.

To make these experimental details explicit and ensure the reproducibility of our work, we have significantly improved the descriptions in the "Methods" section of the revised manuscript, which is presented below.

In page 18 of the revised manuscript:

“Subsequently, we employed magnetron sputtering with molybdenum as the target material to deposit a molybdenum film. To ensure uniform deposition on 4-inch wafers, we controlled the deposition rate between 0.1-0.3 Å s⁻¹ by adjusting the sputtering DC power (8 W) and chamber pressure (0.7-0.8 Pa).”

In page 19 of the revised manuscript:

Secondly, atomic layer deposition was employed to prepare 16 nm HfO₂ or 20 nm Al₂O₃ insulating layers. The HfO₂ film is deposited at a deposition temperature of 250°C using tetrakis (ethylmethylamido) hafnium and H₂O as the precursors. The tetrakis (ethylmethylamido) hafnium precursor was pulsed for 500 ms with a purge time of 5 s. The H₂O precursor was pulsed for 500 ms with a purge time of 8 s. The Al₂O₃ film is deposited at 250 °C using trimethylaluminium and H₂O as the precursors. The trimethylaluminium precursor was pulsed for 500 ms with a purge time of 5 s. The H₂O precursor was pulsed for 500 ms with a purge time of 8 s. Subsequently, photolithography was performed using standard photolithography techniques, followed by process etching employing inductively coupled plasma reactive ion etching (ICP-RIE) under an atmosphere of 80W RF bias power with SF₆/O₂/Ar = 120/40/40 s. c. m. The MoTe₂ film is then transferred to the substrate, the source and drain on the MoTe₂ film of the device are prepared using lithography and electron beam evaporation of Ti/Pd/Au (1/40/20 nm). Finally, the MoTe₂ films is patterned using photolithography and the aforementioned ICP-RIE etching process.

4. Transfer Process: The current fabrication flow relies on transferring the MoTe₂ films. While successful for this demonstration, transfer processes inevitably introduce defects, interface impurities, and yield challenges that are generally incompatible with large-scale CMOS manufacturing. It would be beneficial for the authors to acknowledge these limitations and briefly discuss pathways toward transfer-free (direct growth) integration in the future.

Response: We thank the reviewer for raising the important point regarding the transfer process and its implications for large-scale integration. We fully agree that the transfer of two-dimensional materials can introduce defects, interface impurities, and yield challenges for device yield and uniformity, which are critical considerations for CMOS-compatible manufacturing.

In our current demonstration, the reliance on transfer arises primarily from the challenges of top-gated 2D transistor fabrication using standard nanodevice fabrication processes. It is well known that the atomically flat and chemically clean surface of two-dimensional (2D) materials lacks sufficient nucleation sites for the atomic layer deposition (ALD) of gate dielectrics, making it highly challenging to achieve high-quality, conformal growth directly on 2D material surfaces using standard ALD processes. Therefore, in this work, to ensure device stability, yield, and performance uniformity, we employed a back-gate architecture in which the gate dielectric is pre-deposited on the substrate using well-established and controllable processes. Consequently, a transfer process is necessarily employed in this work, since the high temperatures involved in direct MoTe₂ growth would adversely affect the ALD-deposited gate dielectrics.

We fully agree We fully agree that avoiding transfer processes is essential for improving industrial scalability and compatibility. Consequently, devising direct deposition strategies for high-k gate dielectrics on 2D semiconductors, which are compatible with standard industrial equipment, constitutes a significant challenge. Success in this area would not only streamline manufacturing but also better preserve the intrinsic properties of the 2D channels, and it remains an important objective for the 2D material community.

To make this issue more clearly, we have added new description of such issue in the revised manuscript (in page 16), which is present as below.

“However, the fabrication flow still relies on film transfer to ensure device performance and process yield. Owing to the absence of dangling bonds on two-dimensional materials’ surfaces, conventional atomic layer deposition is nucleation-limited and struggles to form uniform and conformal high-quality gate dielectrics, thereby hindering direct top-gate integration and necessarily need to employ a transfer process of 2D materials in this work. The development of transfer-free, industry-compatible dielectric deposition and in situ integration strategies is expected to simplify processing and facilitate the scalable implementation of two-dimensional CMOS

technologies.”

5. Dielectric Integration: The authors utilize a local back-gate configuration, likely to preserve the channel quality. Top-gate integration is typically preferred for high-density scaling but is often hindered by damage or unintended doping during the Atomic Layer Deposition (ALD) of high-k dielectrics. I suggest the authors discuss potential strategies to solve or mitigate the ALD-induced doping/damage on the MoTe₂ channel, which would be a necessary step for future top-gate MSIC implementation.

Response: We thank the reviewer for raising the important point regarding top-gate integration and the potential impact of ALD on MoTe₂ channel damage or doping. Top-gate architectures are essential for achieving high-density device scaling and large-scale integration; however, the direct deposition of high-k dielectrics on two-dimensional semiconductors presents intrinsic interfacial challenges. Owing to the absence of dangling bonds and reactive surface sites, ALD on two-dimensional materials is typically nucleation-limited, making it difficult to form continuous and high-quality dielectric films and often leading to a high density of interface or body defects of the gate dielectrics. These abundant defects can induce electrostatic doping to the channel, thereby affecting carrier transport characteristics and device uniformity.

To address these issues, we consider interface engineering to be an effective strategy for achieving high-quality top-gate integration. Specifically, introducing an ultrathin and uniform seed layer on the MoTe₂ surface can provide stable nucleation sites, thereby promoting continuous high-k dielectric growth and reducing interface defect density. Furthermore, appropriate surface modification and interfacial treatments can improve dielectric adhesion and interfacial integrity, mitigating unintended doping. [ref 1, ref 9] Together, these approaches offer a viable pathway toward high-quality top-gated MoTe₂ devices while preserving channel integrity.

To make this issue more clearly, we have added new description of such issue in the revised manuscript (in page 9), which is present as below.

“Although a local back-gate configuration is adopted in this study to preserve channel quality, top-gate architectures remain important for achieving high-density integration in two-dimensional electronics. However, the direct deposition of high-k dielectrics on two-dimensional semiconductors presents intrinsic interfacial challenges. Owing to the absence of dangling bonds, ALD growth is nucleation-limited and can introduce interface defects, thereby degrading carrier transport and device uniformity. Interface

engineering strategies, such as the introduction of an ultrathin seed layer, appropriate surface modification, and interfacial treatments,^{1,9} can promote continuous dielectric growth and reduce interfacial defect density, providing a feasible pathway toward high-quality top-gated device integration.”

Response to Reviewer #4

This manuscript reports the growth of 4-inch, wafer-scale 2D MoTe₂ thin films via the optimization of the tellurium source supply. Using these as-prepared thin films, the authors successfully demonstrated the fabrication of a 140-transistor full adder, which illustrates the potential of their material for 2D CMOS circuit integration. I recommend accepting this manuscript, provided that the authors appropriately address the following comments.

Response: We sincerely thank the reviewer for the thorough and constructive comments. We have carefully revised the manuscript to address each point in detail. Below are our itemized responses.

1.Regarding the novelty of the growth method: The manuscript highlights a precursor engineering strategy. However, the authors should more clearly delineate the fundamental advancements of their method compared to the early CVD growth of 2H-MoTe₂ reported a decade ago (J. Am. Chem. Soc. 2015, 137, 37, 11892-11895) and subsequent studies. The current description suggests modifications rather than a paradigm shift. A more detailed discussion of the specific innovations and their impact would be beneficial.

Response: We thank the reviewer for pointing out these relevant references and offering us the opportunity to clarify the key advances of our work. We fully acknowledge that substantial progress has been made in previous studies on MoTe₂ synthesis, which have laid an important foundation for the field. However, the objective of this study extends beyond optimizing the performance of individual devices and instead targets the realization of medium-scale integrated circuits. To this end, two critical materials challenges remain: achieving wafer-scale growth of highly uniform MoTe₂ films, particularly at the 4-inch scale or larger that could be compatible in industrial facilities, and maintaining precise control over the layer number (especially

down to only 3 layers) while preserving such large-area uniformity. These factors directly govern device-to-device consistency and integration feasibility, and thus constitute key bottlenecks for scalable p-type two-dimensional electronics.

To address these challenges, the present work introduces two fundamental advances in materials growth, which together enable the realization of medium-scale integrated circuits, as detailed below.

1) **Achievement of 4-inch wafer-scale uniformity.** Just as described in our manuscript, While previous reports have demonstrated MoTe₂ growth, the maximum wafer size has been limited to approximately 1 inch or below 2 cm. [ref 17, ref 27 (J. Am. Chem. Soc. 2015, 137, 37, 11892-11895)]. For practical industrial adoption, 2D semiconductors must be scalable to substrate sizes of 100 mm (4 inches) or larger to be compatible with industry-standard high-throughput fabrication tools. The work cited by the reviewer, while valuable, does not meet this scalability requirement.

In our manuscript, we introduce a sustained-release tellurium precursor engineering strategy that could provide slow, precise and continuous tellurium vapor during growth process, which enables the uniform growth of 2H-MoTe₂ across a full 4-inch wafer. This level of homogeneity is essential for circuit integration and has allowed us to demonstrate the first medium-scale integrated circuit comprising over 100 transistors based on a p-type 2D semiconductor, which is an achievement not yet reported elsewhere.

2) **Precise thickness control down to 3 layers.** Scaling the semiconductor body thickness is critical for enhancing gate control and device performance. In this work, we achieve linear thickness tuning from 3 to 20 layers by controlling the Mo precursor thickness. This represents the thinnest wafer-scale MoTe₂ reported to date (3 layers) and delivers a marked improvement in device performance, including an on/off ratio of $\sim 1.29 \times 10^5$, nearly an order of magnitude higher than previously achieved values ($\sim 10^4$).

In summary, the key innovations of this work are the scaling of uniform 2H-MoTe₂ to a 4-inch wafer and the precise thinning of the channel to 3 layers, which together enable the first demonstration of a p-type medium-scale integrated circuit. Taken

together, these results demonstrate that our work achieves substantive advances over previous CVD methods in terms of both large-area uniformity and electrical performance, highlighting the fundamental improvements enabled by our precursor and growth engineering strategies, rather than representing a simple modification of existing approaches.

To make this issue more clearly, we have added new description of such issue in the revised manuscript (in page 3), which is present as below.

“In summary, the key advances of this material growth are the scaling of uniform 2H-MoTe₂ to a 4-inch wafer and the precise thinning of the channel down to 3 layers. Together, these innovations enable the first realization of medium-scale integration, that is, with more than 100 devices, of p-type 2D semiconductors with significantly improved device performance.”

2.Regarding the evidence for uniformity and quality: The claim of high quality and uniformity across the 4-inch wafer is central to this work. To more robustly substantiate this claim, the authors should provide structural characterization, such as selected area electron diffraction (SAED) patterns, from multiple, well-defined locations across the wafer. This data would serve as crucial evidence to support the asserted uniformity.

Response: We sincerely thank the reviewer for this insightful comment regarding the need for more comprehensive wafer-scale characterisation. In response, we have performed additional measurements to more rigorously evaluate the film quality and uniformity across the 4-inch wafer.

As suggested, we have supplemented SAED images (as shown in Fig. S7) at multiple locations, which clearly demonstrate the crystallinity of the film. Collectively, these new datasets provide stronger evidence for the high crystallinity and wafer-level uniformity of the MoTe₂ films.

We have added the corresponding figures and discussion in the revised manuscript to address the reviewer’s concern more thoroughly, which are present as below.

In page 9 of the revised Supporting information:

Figure S7. The figures show SAED patterns selected from different regions of MoTe₂ thin films grown on 4-inch wafers. Scale bar 5 nm.

In page 6 of the revised manuscript:

“To further assess the structural uniformity and crystallinity of the films at the wafer scale, additional transmission electron microscopy (TEM) characterizations were performed at multiple spatial locations across the MoTe₂ films, and the corresponding selected-area electron diffraction (SAED) patterns were obtained (Fig. S7). The SAED patterns collected from different regions exhibit well-defined and consistent diffraction spots, indicating good crystallinity and a uniform crystal structure across the wafer.”

3.Regarding the device performance: The authors report a field-effect mobility of ~ 7 cm²/V·s. This value is lower than what has been previously achieved for MoTe₂ FETs (often in the range of 10-50 cm²/V·s), which creates a potential conflict with the claim

of “high-quality” material. Furthermore, the recent report of 2D β -Bi₂O₃ with a hole mobility of 136.6 cm²/V·s (Nat. Mater. 24, 688–697, 2025) sets a new benchmark for p-type 2D semiconductors. To enhance the potential of the 2H MoTe₂ synthesized by authors, it is critical to improve the mobility of single FET before pushing to integrated devices.

Response: We thank the reviewer for the important comments regarding the reported field-effect mobility and its relevance to the assessment of material quality and integration potential. We acknowledge that the average mobility reported in this work (~ 7 cm²/V·s) is lower than some previously reported peak values for MoTe₂ field-effect transistors (typically 10–50 cm²/V·s). However, this difference should be interpreted in the context of device statistics, material form, and fabrication objectives.

First, it is important to note that many of the higher mobilities reported in the literature correspond to best-performing devices (or the “champion” device), rather than statistically representative datasets. In our wafer-scale device arrays, we likewise observe individual devices with mobilities exceeding 10 cm²/V·s (~ 11 cm²/V·s), falling within the previously reported range. The mobility values presented in this work represent statistically averaged results obtained from a large number of devices across a 4-inch wafer, which more accurately reflect device performance under realistic integration conditions.

Second, most prior high-mobility MoTe₂ transistors were fabricated on mechanically exfoliated flakes or single-domain materials, which can be effectively regarded as single crystalline within the channel region. In contrast, the present work employs wafer-scale polycrystalline MoTe₂ films. The presence of grain boundaries in such films is known to introduce additional carrier scattering, which can reduce mobility compared with single-domain or single-crystal devices. Therefore, a moderate reduction in mobility for wafer-scale polycrystalline films is expected and consistent with established transport physics in two-dimensional semiconductors.

Third, the extracted mobility in our devices does not directly reflect the intrinsic transport capability of the MoTe₂ channel. To ensure compatibility with standard semiconductor manufacturing processes and enable wafer-scale batch fabrication, we adopted scalable and conventional contact schemes rather than highly specialized contact engineering approaches that are challenging to be scalable and highly controllable, e.g., the transferred van de Waals contacts. As a result, the contact resistance in the present devices remains relatively high (extracted by Y function methods shown in supplementary note 1, shown in Fig. S15), which limits the measured

channel current and leads to an underestimation of the apparent field-effect mobility. When the influence of contact resistance is decoupled, the effective channel mobility could be higher. While some previously reported high-mobility devices benefit from unconventional contact strategies, such approaches often face challenges in reproducibility, scalability, and integration.

As noted in the references cited by the reviewer (Nat. Mater. 24, 688-697, 2025), the reported carrier mobilities were primarily obtained from discrete devices fabricated within single crystalline domains of approximately 40 μm , which are intended to evaluate the intrinsic transport potential of the material in locally ideal regions. In contrast, the present work focuses on achieving statistically mobility distribution and device-to-device consistency across an entire 4-inch wafer, extending the evaluation scale from isolated domains to wafer-scale uniform films. The realization of this level of material uniformity and integration scalability highlights the practical value of our approach for large-area and application-oriented electronics.

In future work, we will focus on developing low-resistance contact strategies that remain compatible with semiconductor manufacturing processes, with the goal of further enhancing the extracted carrier mobility from transfer characteristics while preserving scalability and integration feasibility.

In page 17 of the revised Supporting information:

Figure S15. The Y function method analysis. **a** Calculated Y function for the devices shown in a as a function of V_{gs} . **b** The R_{C} of devices with different L_{ch} ($W_{\text{ch}}=100\mu\text{m}$).

Response to Reviewer #5

Response: We sincerely thank the reviewer for their positive assessment of our work. We greatly appreciate their constructive evaluation of the manuscript and the recommendation for publication, which is highly valuable to us.

Response to the Reviewers' comments:

Response to Reviewer #3

I would like to thank the authors for their hard work and the comprehensive point-by-point response to the reviewers' comments. The revised manuscript has been substantially improved. Before final acceptance, I suggest the following minor refinements: The author observes a decrease in the ON-state current (I_{on}) after exposing the device to air. However, it is widely reported in the literature that the surface oxidation of $MoTe_2$ typically results in the formation of MoO_x , which induces a strong p-doping effect. Theoretically, this p-doping should lead to an increase in the I_{on} for p-type or ambipolar $MoTe_2$ devices. The author should provide a detailed explanation or additional experimental evidence to clarify why the current decreased in this specific case, addressing whether factors such as increased scattering, contact resistance degradation, or moisture-induced trapping are dominant.

Response: We thank the reviewer for this important point. Surface oxidation of $MoTe_2$ can indeed form MoO_x and induce p-type doping, which in principle may increase I_{on} for p-type or ambipolar devices [ref.18 and ref. 49]. However, previous studies indicate that the response of $MoTe_2$ to ambient exposure is more complex than oxidation alone, as adsorption of O_2 and H_2O at surface defects can significantly influence carrier transport [Adv. Mater. 2018, 30, 1706995].

In our devices, after 7 days of air exposure the transistors remain fully operational but show a slight reduction in I_{on} with minimal threshold-voltage shift. This behavior suggests that air-induced adsorption may introduce additional carrier scattering, moisture-related trapping, which will outweigh the beneficial effect of oxidation-induced p-doping. Further studies will be conducted to distinguish channel- and contact-limited contributions.